# Average Sensitivity of Hierarchical $k$-Median Clustering

**Shijie Li**[1]  **Weiqiang He**[1]  **Ruobing Bai**[1]  **Pan Peng**[1]

## Abstract

Hierarchical clustering is a widely used method for unsupervised learning with numerous applications. However, in the application of modern algorithms, the datasets studied are usually large and dynamic. If the hierarchical clustering is sensitive to small perturbations of the dataset, the usability of the algorithm will be greatly reduced. In this paper, we focus on the hierarchical $k$-median clustering problem, which bridges hierarchical and centroid-based clustering while offering theoretical appeal, practical utility, and improved interpretability. We analyze the average sensitivity of algorithms for this problem by measuring the expected change in the output when a random data point is deleted. We propose an efficient algorithm for hierarchical $k$-median clustering and theoretically prove its low average sensitivity and high clustering quality. Additionally, we show that single linkage clustering and a deterministic variant of the CLNSS algorithm exhibit high average sensitivity, making them less stable. Finally, we validate the robustness and effectiveness of our algorithm through experiments.

## 1. Introduction

*Hierarchical clustering*, as discussed in (McQuitty, 1957; Hastie et al., 2009), is one of the most commonly used clustering strategies. It produces a set of nested clusters, organized into a tree-like structure known as a dendrogram. In this structure, each leaf node typically encompasses only a single object, whereas each internal node represents the union of its child nodes. By exploring the dendrogram's multiple levels, one can identify clusters with varying levels of granularity. Consequently, hierarchical clustering is adept at detecting and analyzing complex data structures. Hierarchical clustering has attracted great interest in computer science (Jin et al., 2015; Charikar & Chatziafratis, 2017; Dasgupta, 2016; Moseley & Wang, 2023; Lin et al., 2010) because of its ability to reveal nested patterns in real-world data. Several agglomerative and divisive methods have been proposed and widely used in practice for hierarchical clustering, such as those based on single, complete, average linkage and Ward's method (see Appendix B).

On the other hand, the stability of algorithms is becoming increasingly important in practical applications. If an algorithm is highly sensitive to changes in data points, even minor alterations can significantly impact the clustering results, leading analysts to make incorrect decisions based on unreliable outcomes, which can, in turn, result in higher costs for error correction. At the same time, such instability may inadvertently expose sensitive information, as small perturbations in data could highlight individual contributions or reveal underlying patterns that compromise privacy. Furthermore, unstable algorithms may erode trust in the analysis process, making their adoption in critical applications more challenging.

It is thus crucial to understand if a hierarchical clustering algorithm is stable. Indeed, several studies have found that many hierarchical clustering algorithms, such as those using single linkage or complete linkage, are sensitive to small changes in the data (Balcan et al., 2014; Cheng et al., 2019; Eriksson et al., 2011). This sensitivity can cause the dendrogram structure to change dramatically with slight variations in the data. However, most of these works focus only on *adversarial* perturbations or attacks on the input and/or lack theoretical rigor.

In this paper, we investigate the question of the stability of hierarchical clustering by adopting the concept of *average sensitivity*, as introduced by (Peng & Yoshida, 2020; Varma & Yoshida, 2021) that provides a theoretical framework for measuring the stability of the output of hierarchical clustering when data points are *randomly* removed from the original dataset. Roughly speaking, for a (deterministic) algorithm that outputs $k$ clusters, its average sensitivity is the expected size of the symmetric difference of the output clusters before and after we randomly remove a few points. One can generalize this notion to random algorithms for hierarchical clustering. This notion effectively captures the scenario where adversarial perturbations of the input are

---

[1]School of Computer Science and Technology, University of Science and Technology of China, Hefei, China. Correspondence to: Pan Peng <ppeng@ustc.edu.cn>.

*Proceedings of the $42^{nd}$ International Conference on Machine Learning*, Vancouver, Canada. PMLR 267, 2025. Copyright 2025 by the author(s).

rare, but random noise is still commonly expected, such as in data collection with a few missing items or in a sensor network with a few sensors out of operation.

We study the fundamental *hierarchical Euclidean k-median clustering problem*, which bridges hierarchical clustering (providing multiscale structure) and centroid-based clustering (optimizing a global objective). This makes it both *theoretically appealing* and practically useful. While linkage-based hierarchical methods are widely used, they lack a well-defined objective function, limiting their theoretical analysis. Moreover, unlike linkage methods that merge clusters based on pairwise distances, hierarchical $k$-median clustering maintains explicit cluster centers, making it more interpretable and well-suited for applications like data summarization or facility location problems.

Specifically, given a set of points $P \subseteq \mathbb{R}^d$, the objective of the hierarchical $k$-median problem (Lin et al., 2010), is to return an ordered set of centers $\{c_1, \ldots, c_n\}$, and a *nested* sequence of cluster sets $\mathcal{P}_1, \ldots, \mathcal{P}_n$ of $P$ so that the maximum over all $1 \le k \le n$ of the ratio of $\sum_{i=1}^{k} \sum_{p \in C_i} \|p - c_i\|_2$ to the optimum unconstrained $k$-median cost is minimized, where $C_i = \{c_1, \ldots, c_i\}$ is the set of centers and $\mathcal{P}_k = \{P_1, \ldots, P_k\}$ is a $k$-partition. Here, $\mathcal{P}_i$ is obtained by splitting a particular cluster $P_j$ from $\mathcal{P}_{i-1} = \{P_1, \ldots, P_{i-1}\}$, where $j \in [i-1]$, resulting in $P_j$ being divided into $\{P_j \setminus P_i, P_i\}$. Consequently, $\mathcal{P}_i$ becomes $\{P_1, \ldots, P_j \setminus P_i, \ldots, P_i\}$. Each $P_i$ is a subset of a part from the partition $\mathcal{P}_{i-1}$.

Several papers have proposed algorithms to address this problem (Lin et al., 2010; Lattanzi et al., 2020). In particular, Cohen-Addad, Lattanzi, Norouzi-Fard, Sohler and Svensson (Cohen-Addad et al., 2021) recently gave a parallel algorithm, which in turn is based on a sequential algorithm that we refer to as the CLNSS algorithm, with small space and round complexity for the above problem that outputs a hierarchical clustering such that for any fixed $k$, the $k$-median cost of the solution is at most an $O(\min\{d, \log n\} \log \Lambda)$ factor larger in expectation than that of an optimal solution, where $\Lambda$ is the ratio between the maximum and minimum distance of two points in the input dataset.

However, it remains unclear whether the CLNSS algorithm and other hierarchical clustering algorithms in Euclidean space, has small average sensitivity (i.e., good stability).

## 1.1. Our Results

In this work, we study the stability of hierarchical $k$-median clustering algorithms for Euclidean space under random perturbations using the concept of average sensitivity. Our main contribution is summarized as follows.

(1) We give a new algorithm for hierarchical $k$-median clustering by incorporating the exponential mechanism that was commonly used in differential privacy with the CLNSS algorithm. Specifically, the CLNSS algorithm first constructs a 2-hierarchically well separated tree (2-RHST; see Appendix C.2) and then invokes a greedy algorithm to selection the centers according the tree distance. To ensure stability, we utilize the exponential mechanism by iteratively selecting the centers according to a probability distribution based on the cost of a point as a potential center. We provide a theoretical guarantee on the performance of our algorithm by proving that our algorithm exhibits small average sensitivity while preserving the utility guarantee (Section 3).

(2) We show that several agglomerative clustering algorithms, including single linkage clustering and a variant of the CLNSS algorithm have large average sensitivity (Section 4). That is, we identify specific datasets where these algorithms demonstrate poor stability. This implies that such algorithms are unstable under random perturbation of the data.

(3) In Section 5, we evaluate our algorithm on multiple datasets. Our results show that our algorithm achieves low average sensitivity while maintaining good approximation guarantees. Additionally, we validate the instability of classic hierarchical clustering algorithms using both synthetic and real datasets. We also observe that single linkage exhibits good robustness on the selected real-world dataset (albeit at the cost of a high $k$-median cost.). To explain this phenomenon, we show – both theoretically and experimentally (Appendix G) – that single linkage achieves low sensitivity on datasets with strong cluster structures.

**Comparison to the work (Hara et al., 2024)** Hara et al. (2024) also examines the average sensitivity of hierarchical clustering. However, the clustering criteria in our study differ significantly from theirs, as we offer much stronger theoretical guarantees. Several key distinctions set our approach apart. First, while their method relies on sparsest cut and Dasgupta cost (Dasgupta, 2016) to construct the hierarchical clustering, our study employs a hierarchical $k$-median clustering approach. Second, although they also leverage the exponential mechanism, they do not provide an approximation guarantee, leaving the effectiveness of their method without theoretical support.

## 1.2. Other Related Work

**Robust or Stable Hierarchical Clustering.** There has been research on robust/stable hierarchical clustering (Balcan et al., 2014; Cheng et al., 2019; Eriksson et al., 2011), but these studies aim to achieve stable clustering results by identifying input outliers. They consider the impact of outliers on clustering, it means that they potentially classify input points as "good points" or "bad points". In fact, even without outliers, the clustering results are not necessarily stable. Any change in data points may cause drastic changes in

hierarchical clustering results. Therefore, our work follows a more natural consideration, and we use a more natural robustness standard.

**Average Sensitivity of Algorithms.** The concept of average sensitivity, pivotal in assessing the stability of algorithms against minor perturbations in input data, was first introduced by Murai & Yoshida (2019). They studied the stability of network centralities. Varma & Yoshida (2021) apply average sensitivity more widely to graph problem analysis, such as maximum matching, minimum s-t cut, etc. Kumabe & Yoshida (2022) extended the study of average sensitivity to dynamic programming algorithms. Peng & Yoshida (2020) analyzed the average sensitivity of $k$-way spectral clustering to evaluate its reliability and efficiency. The average sensitivity of $k$-means++ and coresets which is studied by Yoshida & Ito (2022) further explored these concepts. The application of average sensitivity to decision tree problems was studied by Hara & Yoshida (2022).

**Differential Privacy.** The concept of *differential privacy (DP)* is similar to that of average sensitivity. DP (Dwork et al., 2006) which developed by Dwork et al. is a data privacy protection standard that states that if given two adjacent databases, an differential private algorithm produces statistically indistinguishable outputs. DP clustering algorithms have been widely studied and designed by (Su et al., 2016; Huang & Liu, 2018; Ghazi et al., 2020; Cohen-Addad et al., 2022; Imola et al., 2023), such as $k$-means, $k$-median, correlation clustering and hierarchical clustering.

Note that the difference is that DP focuses on worst-case sensitivity rather than the average-case sensitivity. For example, as long as the algorithm is very sensitive to a certain input, we consider the algorithm to be bad under the concept of DP. In addition, due to the requirements of the definition of DP, the total variation distance of the results of two adjacent inputs must be small, but the earth mover's distance of these two results is smaller than the total variation distance, which means that even if an algorithm is not DP, its average sensitivity may be small. Importantly, it is known that if an algorithm is $\beta$-DP (see e.g., (Dwork et al., 2006)), then its average sensitivity is at most $\beta$ (Varma & Yoshida, 2021). Despite these differences, DP and average sensitivity are both measures of algorithmic robustness, and there are meaningful connections between them.

We provide more related work on statistically robust clustering in Appendix A.

## 2. Preliminaries

**Distances and $k$-Median Cost** For two points $p$ and $q$ in Euclidean space, the Euclidean distance between them is defined as $\text{DIST}(p, q) = \|p - q\|$. Given a point $p$ and a set $C \subseteq \mathbb{R}^d$, we define $\text{DIST}(p, C) = \min_{c \in C} \|p - c\|$, as the minimum distance from $p$ to any point in $C$.

Given a set of points $P \subseteq \mathbb{R}^d$, a set of centers $C = \{c_1, \ldots, c_k\}$, the *$k$-median cost* is defined as

$$\text{COST}(P, C) = \sum_{p \in P} \text{DIST}(p, C) = \sum_{p \in P} \min_{c \in C} \|p - c\|,$$

The points in $C$ are referred to as *cluster centers*.

For two partitions $\mathcal{P}_1$ and $\mathcal{P}_2$, we say that $\mathcal{P}_1$ is nested in $\mathcal{P}_2$ if $\mathcal{P}_2$ can be obtained from $\mathcal{P}_1$ by merging two or more parts of $\mathcal{P}_1$.

**Average Sensitivity** For hierarchical clustering, the input is a point set $P = \{p_1, p_2, \ldots, p_n\}$, and we want to measure the average sensitivity of the algorithm after randomly deleting a point. We let $P^{(i)} = \{p_1, \ldots, p_{i-1}, p_{i+1}, \ldots, p_n\}$ denote the dataset in which the $i$-th point is deleted.

Consider a deterministic algorithm $\mathcal{A}$ that takes as input a point set $P$ and a parameter $k$ and outputs $\mathcal{P}_k = \{P_1, \ldots, P_k\}$. Let $\mathcal{P}_k^{(i)} = \{P_1^{(i)}, \ldots, P_k^{(i)}\}$ denote the output clustering when $\mathcal{A}$ takes as input $P^{(i)}$ and $k$. The average sensitivity of $\mathcal{A}$ on $P$ is given by

$$\beta(\mathcal{A}, P) = \frac{1}{n} \sum_{i=1}^{n} |\mathcal{P}_k \triangle \mathcal{P}_k^{(i)}| = \frac{1}{n} \sum_{i=1}^{n} \min_{\pi} \sum_{j=1}^{k} \left| P_j \triangle P_{\pi(j)}^{(i)} \right|,$$
(1)

where $\pi : [k] \to [k]$ is a bijection, establishing a correspondence between the clusters in $\mathcal{P}_k$ and $\mathcal{P}_k^{(i)}$ based on the symmetric difference of sets. Specifically, for any two sets $X$ and $Y$, the symmetric difference is defined as $X \triangle Y := (X \setminus Y) \cup (Y \setminus X)$.

For a randomized algorithm, we often measure the average sensitivity by the distance between its output distributions. Specifically, the average sensitivity of a randomized algorithm $\mathcal{A}$ on a dataset $P$ (with respect to the total variation distance) is defined as

$$\beta(\mathcal{A}, P) = \frac{1}{n} \sum_{i=1}^{n} d_{\text{EM}}(\mathcal{A}(P), \mathcal{A}(P^{(i)})), \qquad (2)$$

where $d_{\text{EM}}(\mathcal{A}(X), \mathcal{A}(X^{(i)}))$ represents the Earth Mover's Distance (EMD) between the outputs $\mathcal{A}(X)$ and $\mathcal{A}(X^{(i)})$, with the distance between two outputs measured by the symmetric difference (see Eq. (1)). Specifically, $d_{\text{EM}}(\mathcal{A}(X), \mathcal{A}(X^{(i)}))$ is defined as $\min_{\mathcal{D}}[\mathbb{E}_{(x,y) \sim \mathcal{D}}[x \triangle y]]$, where $\mathcal{D}$ denotes a distribution over pairs $(x, y)$ of outputs of $\mathcal{A}$ such that the left and right marginals of $\mathcal{D}$ correspond to $\mathcal{A}(X)$ and $\mathcal{A}(X^{(i)})$, respectively.

**The CLNSS algorithm** Now we describe the CLNSS algorithm. We need the following notion of $\ell$-RHST, which is way of embedding the dataset into a restricted hierarchical structure based on a quadtree.

**Definition 2.1.** *A restricted $l$-hierarchically well-separated tree ($l$-RHST) is a positively weighted rooted tree where all leaves are at the same level, and edges at each level have the same weight, and the length of the edges decreases by a factor of $\ell$ on any root-to-leaf path.*

We will describe in detail on how to build 2-RHST, i.e., algorithm CONSTRUCT2RHST($P$), in Algorithm 4 of Appendix C.2. Given such a tree, let $\mathrm{DIST}_T(p, q)$ denote the shortest path between any two points $p$ and $q$ in a tree $T = (P, E, \omega)$, where $P$ is the point set, with all points $p$ located in the leaf nodes of the tree and at the same level, $E$ is the edge set, and $\omega$ denotes the edge weights, forming the metric space $(P, \mathrm{DIST}_T)$. For a set of points $P$ on a tree and a set of centers $C$ where $|C| = k$, we define $\mathrm{COST}_T(P, C) = \sum\limits_{p \in P} \min_{c \in C} \mathrm{DIST}_T(p, c)$.

The CLNSS algorithm begins by applying a random shift to set $P$, where the shift is uniformly sampled from $[0, \Lambda]^d$. Then, a 2-RHST $T$ is constructed on the shifted dataset, and hierarchical clustering process Algorithm 1 is performed on the 2-RHST tree $T$. See Algorithm 5 in Appendix C.2.

---

**Algorithm 1** (Cohen-Addad et al., 2021) GREEDY ALGORITHM FOR HIERARCHICAL $k$-MEDIAN ON 2-RHST

**Input:** Set of points $P$, cost function $\mathrm{COST}_T$ defined by a 2-RHST $T$

1 Set $S_0 \leftarrow \emptyset$.
2 Set $\mathcal{P}_0 \leftarrow \{P\}$.
3 Label all internal nodes of the $T$ as unlabelled.
4 **for** $t = 1$ *to* $n$ **do**
5      Let $c_t = \arg\min_{x \in P} \mathrm{COST}_T(P, x \cup S_{t-1})$.
6      Label the highest unlabelled ancestor of $c_t$ with $c_t$.
7      Set $S_t$ to be $c_t \cup S_{t-1}$.
8      Define $\mathcal{P}_t$ as the clustering obtained by assigning all points to the cluster centered at their closest labeled ancestor.

**Output:** Return $c_1, \ldots, c_n, \mathcal{P}_1, \ldots, \mathcal{P}_n$

---

We have the following theorem about the CLNSS algorithm.

**Theorem 2.2** ((Cohen-Addad et al., 2021)). *For any $k \in \{1, \ldots, n\}$, let $C^{\star}_{T,k}$ be the center set representing the optimal solution to the $k$-median problem on the 2-RHST $T$, where $T$ is randomly shifted as described above. Then it holds that for any $k \in \{1, \ldots, n\}$,*

*(1) the set $\{c^{\star}_1, \ldots, c^{\star}_k\}$ of the first $k$ centers output by Algorithm 1 is exactly $C^{\star}_{T,k}$,*

*(2) and $\mathbb{E}[\mathrm{COST}(P, C^{\star}_{T,k})] = O(d \cdot \log \Lambda) \cdot \mathrm{OPT}(P, k)$, where $\mathrm{OPT}(P, k)$ represents the optimal cost of the $k$-median problem for the point set $P$.*

## 3. An Algorithm with Low Average Sensitivity

In this section, we present our low-sensitivity algorithm for Hierarchical $k$-Median Clustering and provide a theoretical guarantee of its performance (see Theorem 3.1). The algorithm, detailed in Algorithm 2, incorporates exponential mechanism with Algorithm 1 to ensure low sensitivity.

The algorithm begins with the set $P$ and incrementally constructs the hierarchical clustering from top to bottom. This process unfolds over $n$ iterations, where in each iteration the exponential mechanism (Appendix C.1) is used to select a center point. This selected center then divides the current cluster into two new clusters, driving the hierarchical structure forward.

We first introduce some definition. Let $T$ denote the 2-RHST constructed from the point set $P$. Furthermore, let $T^{(i)}$ represent the 2-RHST derived from the set $P^{(i)}$, which is obtained after the removal of the $i$-th point.

For each $t = 1, \ldots, n$, let $S_{t-1}$ denote the set of centers selected in the first $t - 1$ rounds from the point set $P$, and let $S^{(i)}_{t-1}$ represent the set of centers selected in the first $t - 1$ rounds from the subset $P^{(i)}$, $x \cup S$ to represent $\{x\} \cup S$, where $x$ is a point and $S$ is a set. Let $\bar{x}_t = \arg\min_{p \in P} \mathrm{COST}_T(P, p \cup S_{t-1})$. This optimal center $\bar{x}_t$ is crucial for maintaining the desired clustering properties throughout the hierarchical process. Then, we use the exponential mechanism to select a new center $c_t$, where the parameter $\lambda$ is approximately $\frac{\varepsilon \cdot \mathrm{COST}_T(P, \{\bar{x}_t\} \cup S_{t-1})}{\ln n}$. This iterative process yields a sequence of centers and the resulting clustering.

Compared to the CLNSS algorithm (Algorithm 5), Algorithm 2 has low average sensitivity while ensuring utility. We provide a theoretical guarantee for the algorithm in Theorem 3.1. Recall that $\mathrm{OPT}(P, k)$ represents the optimal cost of the $k$-median problem for the point set $P$.

**Theorem 3.1.** *Given a point set $P$ of size $n$ and a parameter $\varepsilon > 0$, Algorithm 2 provides an approximation for the hierarchical $k$-median clustering problem for any $k \in \{1, \ldots, n\}$. Specifically, it achieves an expected cost of*

$$\mathbb{E}[\mathrm{COST}_T(P, S_k)] \leq O(d \cdot \log \Lambda \cdot (1 + \varepsilon)^k) \cdot \mathrm{OPT}(P, k)$$

*with average sensitivity $O\left(\frac{k \ln n}{\varepsilon}\right)$, for any $k \in \{1, \ldots, n\}$, with probability at least $1 - \frac{k}{n^2}$. And the running time of the Algorithm 2 is $O(dn \log \Lambda + n^3)$.*

By Theorem 3.1, we can easily derive Corollary 3.2. The proof details of Theorem 3.1 and Corollary 3.2 are both deferred to Appendix D.

**Corollary 3.2.** *Considering the setting in Theorem 3.1, we have that*

$$\mathbb{E}\left[\max_k \frac{\mathrm{COST}_T(P, S_k)}{\mathrm{OPT}(P, k) \cdot (1 + \varepsilon)^k}\right] = O\left(d \cdot \log \Lambda \log(dn\Lambda)\right).$$

**Algorithm 2** A LOW-SENSITIVITY ALGORITHM FOR HIERARCHICAL $k$-MEDIAN CLUSTERING

**Input:** Set of points $P$

1  Apply a random shift to each point in $P$, where the shift is uniformly drawn from $[0, \Lambda]^d$.
2  Construct 2-RHST and let $\text{COST}_T$ denote the cost function defined by $T$.
3  Set $S_0 \leftarrow \emptyset$.
4  Set $\mathcal{P}_0 \leftarrow \{P\}$.
5  Label all internal nodes of the RHST as unlabelled.
6  **for** $t = 1$ **to** $n$ **do**
7  $\qquad$ Randomly sample a number $\lambda \in \left[\frac{\varepsilon \cdot \text{COST}_T(P, \{\bar{x}_t\} \cup S_{t-1})}{6 \ln n}, \frac{\varepsilon \cdot \text{COST}_T(P, \{\bar{x}_t\} \cup S_{t-1})}{3 \ln n}\right]$.
8  $\qquad$ Let $c_t = x$, where $x$ is sampled with probability $\propto \exp(-\text{COST}_T(P, x \cup S_{t-1})/\lambda)$.
9  $\qquad$ Label the highest unlabelled ancestor of $c_t$ with $c_t$.
10  $\qquad$ Set $S_t = c_t \cup S_{t-1}$.
11  $\qquad$ Define $\mathcal{P}_i$ as the clustering obtained by assigning all points to the cluster centered at their closest labelled ancestor.

**Output:** Return $c_1, \ldots, c_n, \mathcal{P}_1, \ldots, \mathcal{P}_n$

We sketch the proof of Theorem 3.1 in the following.

### 3.1. Proof of Theorem 3.1: Sensitivity

We demonstrate the average sensitivity by proving it through the average sensitivity at each level of hierarchical clustering. More specifically, when proving the $t$-th layer, we assume that the set of centers chosen in the previous $t - 1$ layers is fixed (denoted as $S_{t-1}$), and then calculate the average sensitivity of the $t$-th layer under this condition.

We first introduce some definitions.

Let $T(X, y) = \text{COST}_T(X, y \cup S_{t-1})$ (resp. $T^{(i)}(X, y) = \text{COST}_{T^{(i)}}(X, y \cup S_{t-1})$). Here, $X$ represents a set and $y$ represents a point such that $y \in X$ and $X \subseteq \mathbb{R}^d$. Define $A^{(i)}(x)$, $B$, and $B^{(i)}$ as follows:

$$A^{(i)}(x) = \left| \frac{\exp\left(\frac{-T(P,x)}{\lambda}\right)}{\sum\limits_{p \in P} \exp\left(\frac{-T(P,x)}{\lambda}\right)} - \frac{\exp\left(\frac{-T^{(i)}(P^{(i)},x)}{\lambda}\right)}{\sum\limits_{p \in P^{(i)}} \exp\left(\frac{-T^{(i)}(P^{(i)},x)}{\lambda}\right)} \right|$$

$$B = \frac{\varepsilon \cdot T(P, \bar{x}_t)}{6 \ln n}, \quad B^{(i)} = \frac{\varepsilon \cdot T^{(i)}\left(P^{(i)}, \bar{x}_t^{(i)}\right)}{6 \ln(n-1)}.$$

We first present the results of the average sensitivity analysis for the Algorithm 2.

**Lemma 3.3** (Sensitivity). *Let* ALG *represent our algorithm described in Algorithm 2. The average sensitivity of the clustering produced by* ALG, *where* ALG *generates the set of $k$ centers $S_k$ and a partition $\mathcal{P}_k = \{P_1, \ldots, P_k\}$ from*

the input point set $P$, and computes the clustering on $P^{(i)}$ *(obtained by uniformly and randomly deleting one point from $P$) with the resulting centers $S_k^{(i)}$ and partition $\mathcal{P}_k^{(i)}$, is:*

$$\frac{1}{n} \sum_{i=1}^{n} d_{\text{EM}}(\text{ALG}(P, S_k), \text{ALG}(P^{(i)}, S_k^{(i)}))$$

$$= O\left(\frac{k \ln n}{\varepsilon}\right).$$

In order to prove Lemma 3.3, we first show the following:

**Lemma 3.4.** *For Algorithm 2, given a set of points $P$ and $P^{(i)}$, if $S_{t-1}^{(i)} \equiv S_{t-1}$, that is, if the first $t - 1$ centers are the same, then $\frac{1}{n}\sum_{i=1}^{n} d_{\text{TV}}(c_t, c_t^{(i)}) = O(\frac{\ln n}{\varepsilon \cdot n})$, where $d_{\text{TV}}(c_t, c_t^{(i)})$ represents the Total Variation (TV) distance between the distributions of the algorithm's selected $t$-th center $c_t$ and $c_t^{(i)}$.*

Without loss of generality, we assume that the $t$-th center is currently selected, and the centers selected in the previous set $S_{t-1}$ remain consistent, meaning $S_{t-1}^{(i)} = S_{t-1}$. At this stage, we apply Lemma D.1 to bound $d_{\text{TV}}(\text{ALG}(P, S_t), \text{ALG}(P^{(i)}, S_t^{(i)}))$ by $d_{\text{TV}}\left(\text{ALG}(P)|_{\lambda=\hat{\lambda}}, \text{ALG}(P^{(i)})|_{\lambda^i=\hat{\lambda}}\right)$, which represents the expected difference of the distribution under the same choice of $\lambda$. Given that $S_{t-1}^{(i)} = S_{t-1}$, the latter is bounded by $\sum\limits_{x \in P^{(i)}} A^{(i)}(x)$. Then by letting $M = 1$ (an upper bound on the total variation distance), and $\alpha = 1$ in Lemma D.1, we obtain

$$\frac{1}{n}\sum_{i=1}^{n} d_{\text{TV}}(c_t, c_t^{(i)}) \leq \frac{1}{Bn} \int_{B}^{2B} \left(\sum_{i=1}^{n} \sum_{x \in P^{(i)}} A^{(i)}(x)\right) d\hat{\lambda}$$

$$+ \frac{2}{n} \cdot \sum_{i=1}^{n} \left|1 - \frac{B^{(i)}}{B}\right|.$$

Now we bound the first term and the second term separately and give some lemmas. First, we note that

$$\sum_{i=1}^{n} \sum_{x \in P^{(i)}} A^{(i)}(x)$$

$$\leq \sum_{i=1}^{n} \sum_{x \in P^{(i)}} \left| \frac{\exp\left(\frac{-T(P,x)}{\lambda}\right)}{\sum\limits_{p \in P} \exp\left(\frac{-T(P,p)}{\lambda}\right)} - \frac{\exp\left(\frac{-T^{(i)}(P^{(i)},x)}{\lambda}\right)}{\sum\limits_{p \in P} \exp\left(\frac{-T(P,p)}{\lambda}\right)} \right|$$

$$+ \sum_{i=1}^{n} \sum_{x \in P^{(i)}} \left| \frac{\exp\left(\frac{-T^{(i)}(P^{(i)},x)}{\lambda}\right)}{\sum\limits_{p \in P} \exp\left(\frac{-T(P,p)}{\lambda}\right)} \right|$$

$$-\frac{\exp\left(\frac{-T^{(i)}(P^{(i)},x)}{\lambda}\right)}{\sum\limits_{p\in P^{(i)}}\exp\left(\frac{-T^{(i)}(P^{(i)},p)}{\lambda}\right)}\Bigg|$$

$$:=(I)+(II).$$

We bound (I) using Lemma 3.5, bound (II) using Lemma 3.6.

**Lemma 3.5.** *It holds that*

$$(I)\leq O\left(\frac{\ln n}{\varepsilon}\right).$$

*Proof.* By definition of (I), we have

$$(I)=\sum_{i=1}^{n}\sum_{x\in P^{(i)}}\frac{\exp\left(\frac{-T(P,x)}{\lambda}\right)\cdot\left|1-\exp\left(\frac{T(x^{(i)},x)}{\lambda}\right)\right|}{\sum\limits_{p\in P}\exp\left(\frac{-T(P,p)}{\lambda}\right)}.$$

Note that

$$\left|1-\exp\left(\frac{T(x^{(i)},x)}{\lambda}\right)\right|\leq\frac{(e-1)}{\lambda}\cdot T(x^{(i)},x),$$

as $e^{x}-1\leq(e-1)x$, for $x\in[0,1]$. Thus,

$$(I)\leq\sum_{i=1}^{n}\sum_{x\in P^{(i)}}\frac{(e-1)}{\lambda}\cdot\frac{\exp\left(\frac{-T(P,x)}{\lambda}\right)\cdot T(x^{(i)},x)}{\sum\limits_{p\in P}\exp\left(\frac{-T(P,p)}{\lambda}\right)}$$

$$\leq\sum_{i=1}^{n}\sum_{x\in P}\frac{(e-1)}{\lambda}\cdot\frac{\exp\left(\frac{-T(P,x)}{\lambda}\right)\cdot T(x^{(i)},x)}{\sum\limits_{p\in P}\exp\left(\frac{-T(P,p)}{\lambda}\right)}$$

$$\leq O\left(\frac{1}{\lambda}\cdot T(P,\bar{x}_{t})\right)\leq O\left(\frac{\ln n}{\varepsilon}\right).$$

Here, the last inequality follows from the definition of $\lambda$. $\lambda\in\left[\frac{\varepsilon\cdot\text{COST}_{T}(P,\bar{x}_{t}\cup S_{t-1})}{6\ln n},\frac{\varepsilon\cdot\text{COST}_{T}(P,\bar{x}_{t}\cup S_{t-1})}{3\ln n}\right]$, since $\text{COST}_{T}(P,\bar{x}_{t}\cup S_{t-1})$ simplifies to $T(P,\bar{x}_{t})$, we can rewrite this as:

$$\lambda\in\left[\frac{\varepsilon\cdot T(P,\bar{x}_{t})}{6\ln n},\frac{\varepsilon\cdot T(P,\bar{x}_{t})}{3\ln n}\right].$$

$\square$

**Lemma 3.6.** *It holds that*

$$(II)\leq O\left(\frac{\ln n}{\varepsilon}\right).$$

The proof of Lemma 3.6 is deferred to Appendix D.

By these two lemmas, we then derive

$$\sum_{i=1}^{n}\sum_{x\in P^{(i)}}A^{(i)}(x)\leq O\left(\frac{\ln n}{\varepsilon}\right).$$

Next, we aim to bound the term

$$\frac{2}{n}\cdot\sum_{i=1}^{n}\left|1-\frac{B^{(i)}}{B}\right|$$

To this end, we state the following lemma:

**Lemma 3.7.** *It holds that*

$$\frac{2}{n}\cdot\sum_{i=1}^{n}\left|1-\frac{B^{(i)}}{B}\right|\leq O(\frac{1}{n}).$$

The proof of Lemma 3.7 is deferred to Appendix D. We will make use of this bound in the analysis below.

$$\frac{1}{n}\sum_{i=1}^{n}d_{\text{TV}}(c_{t},c_{t}^{(i)})\leq\frac{1}{Bn}\int_{B}^{2B}\left(\sum_{i=1}^{n}\sum_{x\in P^{(i)}}A^{(i)}(x)\right)d\hat{\lambda}$$

$$+\frac{2}{n}\cdot\sum_{i=1}^{n}\left|1-\frac{B^{(i)}}{B}\right|.$$

$$\leq\frac{1}{Bn}\int_{B}^{2B}\left(O\left(\frac{\ln n}{\varepsilon}\right)\right)d\hat{\lambda}+O(\frac{1}{n})$$

$$\leq O\left(\frac{\ln n}{\varepsilon\cdot n}\right).$$

This completes the proof of Lemma 3.4.

Now we are ready to prove Lemma 3.3.

*Proof of Lemma 3.3.* The sensitivity of the clustering after the $j$-th round is given by

$$d_{\text{EM}}(\text{ALG}(P,S_{j}),\text{ALG}(P^{(i)},S_{j}^{(i)}))$$

This can be expressed as the cost of

$$d_{\text{EM}}(\text{ALG}(P,S_{j-1}),\text{ALG}(P^{(i)},S_{j-1}^{(i)}))$$

plus the sum of the costs incurred after selecting the $j$-th center $c_{j}$.

By Lemma 3.4, it holds that

$$\frac{1}{n}\sum_{i=1}^{n}d_{\text{EM}}(\text{ALG}(P,S_{k}),\text{ALG}(P^{(i)},S_{k}^{(i)}))$$

$$\leq\frac{1}{n}\sum_{i=1}^{n}(d_{\text{TV}}(c_{k},c_{k}^{(i)}))\cdot(|\mathcal{P}_{k}\triangle\mathcal{P}_{k}^{(i)}|)$$

$$+ \frac{1}{n} \sum_{i=1}^{n} d_{\text{EM}}(\text{ALG}(P, S_{k-1}), \text{ALG}(P^{(i)}, S_{k-1}^{(i)}))$$

$$\leq O\left(\frac{\ln n}{\varepsilon n}\right) \cdot O(n)$$

$$+ \frac{1}{n} \sum_{i=1}^{n} d_{\text{EM}}(\text{ALG}(P, S_{k-1}), \text{ALG}(P^{(i)}, S_{k-1}^{(i)})),$$

where the first inequality arises from the properties of the Earth Mover's (EM) distance. The lemma then follows by induction.

$\square$

### 3.2. Proof of Theorem 3.1: Utility

Now we prove the utility guarantee of the algorithm.

**Lemma 3.8** (Approximate Ratio). *For the $k$-MEDIAN hierarchical clustering problem, let $S_k$ represent the set of centers for the point set $P$ obtained using Algorithm 2, $|S_k| = k$. Let $C_{T,k}^{\star}$ represent the set of centers for the point set $P$ obtained by the RHST tree-based hierarchical clustering, with $|C_{T,k}^{\star}| = k$. The clusters output by Algorithm 2 satisfy*

$$\mathbb{E}\left[\text{COST}_T(P, S_k)\right] \leq (1+\varepsilon)^k \text{COST}_T(P, C_{T,k}^{\star})$$
$$\leq O(d \cdot \log \Lambda \cdot (1+\varepsilon)^k) \cdot \text{OPT}(P, k),$$

*with probability at least $1 - \frac{k}{n^2}$.*

*Proof.* We prove the lemma by induction.

**Base Case.** We begin by proving the theoretical guarantee for selecting the first center. Let $c_1$ denote the first center chosen by our algorithm, and let $c_1^{\star}$ denote the optimal value under the current conditions, which is the result obtained on 2-RHST. This can be directly established using the exponential mechanism Lemma C.1 with $|\mathcal{R}| = n, r = \ln n^2$ and utility function $u(x, r) = \text{COST}_T(P, x \cup S_{t-1})$. We have the following:

$$\Pr[\text{COST}_T(P, c_1) \leq \text{COST}_T(P, c_1^{\star}) + 3\lambda \cdot \ln n] \geq 1 - \frac{1}{n^2}.$$

Since $\lambda \in \left[\frac{\varepsilon \cdot \text{COST}_T(P, c_1)}{6 \ln n}, \frac{\varepsilon \cdot \text{COST}_T(P, c_1)}{3 \ln n}\right]$, we can get

$$\Pr[\text{COST}_T(P, c_1) \leq (1+\varepsilon)\text{COST}_T(P, c_1^{\star})] \geq 1 - \frac{1}{n^2}.$$

**Inductive Step.** Assume the property holds for $k - 1$ where $k \geq 2$. Consider $C_{T,k-1}^{\star}$, which is the set of the $k-1$ centers chosen by 2-RHST. Suppose that

$$\Pr[\text{COST}_T(P, S_{k-1}) \leq (1+\varepsilon)^{k-1}\text{COST}_T(P, C_{T,k-1}^{\star})]$$
$$\geq 1 - \frac{k-1}{n^2}.$$

We now prove the property for $k$. On the basis of $S_{k-1}$, we can obtain

$$\text{COST}_T(P, c_k \cup S_{k-1}) \leq (1+\varepsilon)\text{COST}_T(P, c_k^{opt} \cup S_{k-1}).$$
(3)

with probability at least $1 - \frac{1}{n^2}$ from Lemma C.1 and the setting of $\lambda$ where $c_k$ is the center choice of Algorithm 2 in $k$-th round and $c_k^{opt}$ is the optimal center choice based on $S_{k-1}$ in $k$-th round.

Because of the meaning of $c_k^{opt}$, it's cost must be less than or equal to $c_i^{\star} \cup S_{t-1}$ for $i \in [k]$. From this, we derive

$$\text{COST}_T(P, c_k^{opt} \cup S_{k-1}) \leq \min_{i \in [k]} \text{COST}_T(P, c_i^{\star} \cup S_{k-1})$$
(4)

By the pigeonhole principle, there must exist a $c_i^{\star}$ for $i \in [k]$ that is a subtree not covered by $S_{k-1}$. Consequently, without loss of generality, we assume that $c_t^{\star}$ is a subtree not covered by $S_{k-1}$. Then, the following equation

$$\text{COST}_T(P \setminus \{\text{rooted at } c_t^{\star}\}, S_{k-1})$$
$$\leq (1+\varepsilon)^{k-1}\text{COST}_T(P \setminus \{\text{rooted at } c_t^{\star}\}, C_{T,k-1}^{\star})$$

holds true. Now we will discuss the classification to show that our conclusion is correct. First, if $c_k^{opt}$ and $c_t^{\star}$ are in the same cluster, then $c_k^{opt}$ and $c_t^{\star}$ are equal. In the second case, if $c_k^{opt}$ is not in the cluster where $c_t^{\star}$ is located, then the cost of the cluster centered on $c_k^{opt}$ is at least equal to that of the cluster centered on $c_t^{\star}$. That is shown in Eq. (4). Thus,

$$\text{COST}_T(P, c_k^{opt} \cup S_{k-1})$$
$$\leq (1+\varepsilon)^{k-1}\text{COST}_T(P, c_t^{\star} \cup C_{T,k-1}^{\star}).$$
(5)

Combining Eq. (3) and Eq. (5), we obtain

$$\text{COST}_T(P, c_k \cup S_{k-1}) \leq (1+\varepsilon)\text{COST}_T(P, c_k^{opt} \cup S_{k-1})$$
$$\leq (1+\varepsilon)^k \text{COST}_T(P, c_t^{\star} \cup C_{T,k-1}^{\star}),$$

which holds with a probability of at least $1 - \frac{k}{n^2}$. Therefore,

$$\Pr[\text{COST}_T(P, S_k) \leq (1+\varepsilon)^k \text{COST}_T(P, C_{T,k}^{\star})] \geq 1 - \frac{k}{n^2}.$$

Then combining Theorem 2.2 we complete the proof. $\square$

## 4. Lower Bounds on the Average Sensitivity of Some Classical Algorithms

In this section, we analyze the average sensitivity of existing hierarchical clustering algorithms through illustrative examples. We first give a lower bound on the sensitivity of the single linkage (see Appendix B) in the worst case.

**Lemma 4.1** (Average sensitivity of Single Linkage). *The average sensitivity of Single Linkage is at least $\Omega(n)$.*

*Proof.* For simplicity, assume the number of clusters $k$ is 2. Let $X = \{x_1, \cdots, x_n\}$ be $n$ points on a line in $\mathbb{R}$, nearly equidistant as shown in Fig. 1. Specifically, suppose $d_1, d_2, \ldots, d_{n-1}$ are the distances between $x_1$ and $x_2$, $x_2$ and $x_3$, $\cdots$, $x_{n-1}$ and $x_n$ respectively, then it follows that

$$d_2 = 1 + \frac{d_1}{n}, d_3 = 1 + \frac{2 \cdot d_1}{n}, \ldots, d_n = 1 + \frac{(n-1) \cdot d_1}{n}.$$

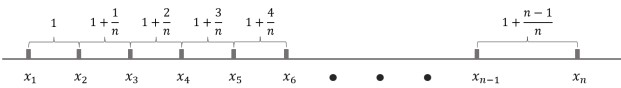

*Figure 1.* $n$ nearly equidistant points on a line.

For $X$, single linkage starts by merging the points with smaller indices first. After deleting $x_i$, single linkage will merge the points to the left and right of $x_i$ separately, and finally merge them. Thus, the symmetric difference of the clustering results of $X$ and $X - \{x_i\}$ by the algorithm is

$$\{\{x_1, \ldots, x_{n-1}\}, \{x_n\}\} \triangle \{\{x_1, \ldots, x_{i-1}\}, \{x_{i+1}, \ldots, x_n\}\}$$

Note that the size of the above set is $n - i$. Thus, the average sensitivity of single linkage in $X$ is $\frac{\sum_{i=1}^{n}(n-i)}{n} = \Omega(n)$. $\quad\square$

While the above results demonstrate that single linkage can have high average sensitivity on certain hard instances, we show in Lemma G.2 that for datasets with strong cluster structure, its sensitivity can be significantly lower.

We also provide the lower bound for the deterministic version of the CLNSS algorithm, whose proof is deferred to Appendix E.

**Lemma 4.2** (Average sensitivity of Algorithm 1). *The average sensitivity of the deterministic version of CLNSS Algorithm (Algorithm 1) is at least $\Omega(n)$.*

## 5. Experiments

We experimentally evaluate the average sensitivity and clustering performance of our proposed algorithm, and compare it against CLNSS algorithm and four well-known linkage algorithm, i.e., single linkage, complete linkage, average linkage, and Ward's method.

All algorithms were implemented in Python 3.10, and the experiments were conducted on a high-performance computing system featuring a 128-core Intel(R) Xeon(R) Platinum 8358 CPU and 504GB of RAM.

**Implementation Changes** Exactly computing the average sensitivity as defined in Eq. (2) is impractical for the experiments, due to high computational cost. Inspired by (Varma & Yoshida, 2021), We exploit the following quantity for

the experiments: $\mathbb{E}_{e \sim E}\left[\mathbb{E}_\pi\left[\left|\mathcal{A}_\pi(P) \triangle \mathcal{A}_\pi(P^{(i)})\right|\right]\right]$. Using this quantity, we measure the expected symmetric difference in the outputs of $\mathcal{A}$ on $P$ and $P^{(i)}$ under *the same random seed* $\pi$. It provides an upper bound the true average sensitivity given in Eq. (2), as shown in (Varma & Yoshida, 2021). Additional experimental setups and results are provided in Appendix F.

**Experiments on synthetic datasets** We first evaluate the sensitivity of our algorithm on synthetic datasets. Using the data from Lemmas 4.1 and 4.2, we observe that our algorithm (with $\varepsilon = 1$, $k = 2$) behaves consistently with our theoretical predictions. Specifically, Single Linkage shows the highest sensitivity on one dataset (Fig. 2a), while CLNSS exhibits high average sensitivity on the other (Fig. 2b). In contrast, our algorithm and some other baseline methods demonstrate low average sensitivity across these datasets. The primary goal of this experiment is to highlight the sensitivity issues in Single Linkage and CLNSS, rather than to emphasize the superiority of our own method.

Next, we test our algorithm on a random regression dataset with 500 points generated with scikit-learn (Pedregosa et al., 2011), varying $k$ and $\varepsilon$. Fig. 3 shows that our algorithm maintains low sensitivity (at most 50). Moreover, larger $\varepsilon$ values lead to smoother sensitivity curves, while smaller $\varepsilon$ values result in greater fluctuations. One reason is that when $\varepsilon$ is close to 0, the algorithm tends to select centers greedily (as in Algorithm 1); when $\varepsilon$ increases (approaching $\infty$), center selection becomes increasingly uniform at random.

**Experiments on real-world datasets** We use datasets from the Scikit-learn library repository (Pedregosa et al., 2011) and the UCI Machine Learning Repository (Asuncion et al., 2007) to evaluate our algorithm. See Table 1 for more detailed description.

**(I) $k$-median cost** We evaluate the $k$-median cost, reflecting the clustering quality, using the criteria defined in Section 2. Here, the cost refers to the $k$-median cost of the $k$-clustering output. The comparison includes our algorithm ($\varepsilon = 1$)[1] and other methods across varying values of $k$, as shown in Figs. 4 and 9. The results indicate that while our algorithm slightly underperforms Ward's method and complete linkage, it outperforms other approaches. Note that single linkage produces significantly higher costs, demonstrating poor performance. Overall, our algorithm achieves clustering costs comparable to other algorithms.

**(II) Average sensitivity** To evaluate an algorithm's average sensitivity, we run it on both the original and perturbed

---

[1]The choice of the $\varepsilon$ parameter largely depends on the specific requirements of the problem. It should be set to balance the trade-off between the approximation ratio and average sensitivity, depending on the desired outcome. For instance, in practice, one could use a geometric search to find an appropriate $\varepsilon$ that satisfies the target accuracy or achieves the desired sensitivity.

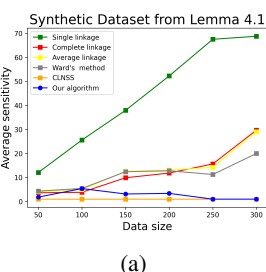
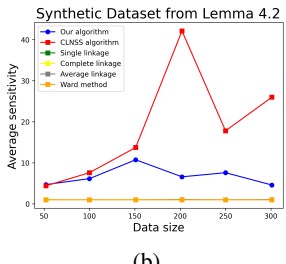
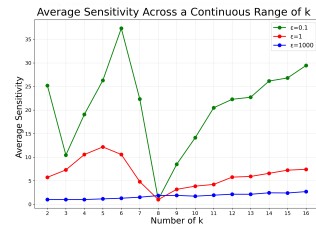

| (a) | (b) |

*Figure 2.* Results for generated datasets described in Lemmas 4.1 and 4.2. The $x$-axis represents the data size, while the $y$-axis represents the average sensitivity.

*Figure 3.* Results for a generated regression dataset with 500 points.

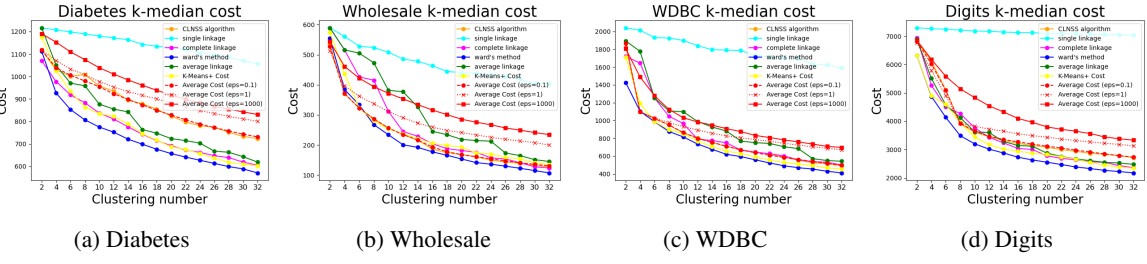

| (a) Diabetes | (b) Wholesale | (c) WDBC | (d) Digits |

*Figure 4.* Results on real-world datasets comparing the $k$-median cost of different algorithms for varying values of $k$

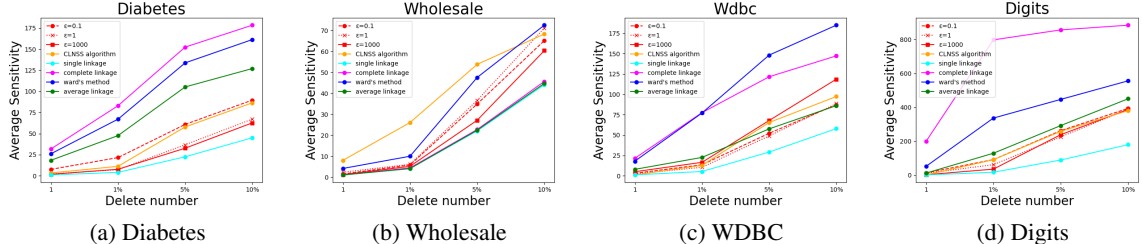

| (a) Diabetes | (b) Wholesale | (c) WDBC | (d) Digits |

*Figure 5.* Results on real-world datasets with $k = 4$. The $x$-axis shows the number of deleted points, and the $y$-axis represents average sensitivity. Our algorithm performs slightly worse than single linkage but surpasses the other algorithms.

datasets (see below) and measure the symmetric difference between the resulting clusterings. This process is repeated 100 times, and the average difference is computed.

Given a dataset, we generate perturbed datasets by removing certain points as follows: (1) For each deletion setting (1 point, 1%, 5%, and 10% deletions), conduct 100 independent trials. (2) In each trial, randomly remove the specified number or percentage of points from the original dataset to create a perturbed version. For a fixed $k$, we evaluate our algorithm with three values of[2] $\varepsilon$: 1, 10, and 1000.

In terms of average sensitivity, Fig. 5 shows that our algorithm ($k = 4$) outperforms traditional linkage-based algorithms and the CLNSS algorithm. (Recall that it achieves

lower clustering costs than average linkage, single linkage, and the CLNSS algorithm, as seen in Figs. 4 and 9.) Furthermore, as $\varepsilon$ increases, the average sensitivity of our algorithm decreases, confirming our theoretical result (Theorem 3.1). We also report that (in Fig. 7) our algorithm consistently maintains lower average sensitivity across different values of $k$, whereas other algorithms (except single linkage) perform well only for certain $k$ values.

While single linkage appears to have lower sensitivity, this seemingly contradicts our lower bound in Lemma 4.1. We believe this is because real-world datasets often have strong clustering structures, unlike worst-case instances. In Lemma G.2, we prove that single linkage has low average sensitivity for well-clusterable datasets and show experimentally (Fig. 10) that many real datasets exhibit similar structures.

---

[2]We also tested $\varepsilon = 10$ and 100, but since center selection probability depends exponentially on $\frac{-1}{\varepsilon}$, the algorithm's behavior remains similar for $\varepsilon = 10, 100, 1000$, with nearly overlapping curves. For clarity, we show only the largest $\varepsilon$.

## Acknowledgments

This work is supported in part by NSFC Grant 62272431 and Innovation Program for Quantum Science and Technology (Grant No. 2021ZD0302901).

## Impact Statement

This paper presents work whose goal is to advance the field of Machine Learning. There are many potential societal consequences of our work, none which we feel must be specifically highlighted here.

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

# A. Further Related Work

**Statistically Robust Clustering.** Several studies evaluate algorithm robustness from a statistical perspective. For instance, The works (Xie & Shekhar, 2019; Henelius et al., 2016) assess robustness using statistical measures. (Xie & Shekhar, 2019) employs Monte Carlo methods to eliminate randomness, reducing "chance patterns" or false positives generated by stochastic processes, thereby enhancing the reliability and robustness of clustering results. Similarly, (Henelius et al., 2016) identifies core clusters by finding the largest set of data points with a co-occurrence probability of at least $1 - \alpha$ across multiple iterations of the same clustering experiment. Additionally, (Monti et al., 2003) introduces consensus clustering, which uses resampling techniques to evaluate cluster stability across repeated iterations. It is important to note that these algorithms evaluate robustness from a statistical perspective, ensuring that their output retains most of the dataset, which differs from our study.

# B. Agglomerative Hierarchical Clustering

A common class of approaches for computing hierarchical clusterings are agglomerative linkage algorithms (Ackermann et al., 2014; Großwendt et al., 2019). A hierarchical clustering can be computed in a bottom-up fashion, where pairs of clusters are merged successively. Agglomerative linkage procedures do exactly that, with the choice of clusters to be merged at every step determined by a linkage function. This linkage function maps all possible pairs of disjoint clusters onto $\mathbb{R}^+$, and the algorithm chooses the pair that minimizes this value.

Before introducing the linkage functions, let's define the distance between points in a cluster. We denote the cost of merging two clusters $A$ and $B$ as $\mathrm{COST}(A, B)$. The linkage functions we are interested in are:

- **Single Linkage.** This linkage function measures the minimum distance between pairs of points from two clusters. The single linkage function is defined as:

$$\mathrm{COST}(A, B) = \min_{(a,b) \in A \times B} \mathrm{DIST}(a, b).$$

  This means that the cost of merging two clusters $A$ and $B$ is the minimum distance between any pair of points $(a, b)$, where $a \in A$ and $b \in B$.

- **Complete Linkage.** This linkage function considers the maximum distance between pairs of points from two clusters. Formally, it is defined as:

$$\mathrm{COST}(A, B) = \max_{(a,b) \in A \times B} \mathrm{DIST}(a, b).$$

  In this case, the cost of merging two clusters $A$ and $B$ is the maximum distance between any pair of points $(a, b)$, where $a \in A$ and $b \in B$.

- **Ward's Method.** This linkage function minimizes the total within-cluster variance. The cost of merging two clusters $A$ and $B$ is defined as the increase in the total within-cluster variance after merging. Formally, it is defined as

$$\mathrm{COST}(A, B) = \sum_{a \in A \cup B} \|a - \mu_{A \cup B}\|^2 - \left( \sum_{a \in A} \|a - \mu_A\|^2 + \sum_{b \in B} \|b - \mu_B\|^2 \right),$$

  where $\mu_A$, $\mu_B$, and $\mu_{A \cup B}$ are the centroids of clusters $A$, $B$, and the merged cluster $A \cup B$, respectively. This means that the cost of merging two clusters is the increase in the sum of squared deviations from the cluster centroids.

- **Average Linkage.** This linkage function measures the average distance between pairs of points from two clusters. Formally, it is defined as:

$$\mathrm{COST}(A, B) = \frac{1}{|A||B|} \sum_{a \in A} \sum_{b \in B} \mathrm{DIST}(a, b).$$

  In this case, the cost of merging two clusters $A$ and $B$ is the average distance between all pairs of points $(a, b)$, where $a \in A$ and $b \in B$.

---

**Algorithm 3** AGGLOMERATIVE ALGORITHM

---

**Input:** $X$ **finite set of input points from** $\mathbb{R}^d$
1 Set $C_{|X|} = \{\{x\} \mid x \in X\}$
2 **for** $i = |X| - 1, \ldots, 1$ **do**
3 $\quad$ find distinct clusters $A, B \in C_{i+1}$ minimizing $\mathrm{COST}(A \cup B)$
4 $\quad$ $C_i = (C_{i+1} \setminus \{A, B\}) \cup \{A \cup B\}$
5 **end**
$\quad$ **Output:** Return clustering $C_1, \ldots, C_{|X|}$

---

The general algorithm flow of the Agglomerative Clustering Algorithm is shown in Algorithm 3. The algorithm flow is the same for these classic linkage hierarchical clustering methods, with the only difference being the calculation of cost during the cluster merging process; the other steps remain identical.

The general algorithm flow of the Agglomerative Clustering Algorithm is illustrated in Algorithm 3. The steps are the same except for the calculation of COST during the cluster merging process.

## C. Supplementary Preliminaries

### C.1. Exponential Mechanism

*The exponential mechanism* (McSherry & Talwar, 2007; Dwork et al., 2014) is a commonly used algorithm in the design of differential privacy, and it is also very useful for designing algorithms with low average sensitivity. Its main idea is to introduce special randomness when selecting candidates, making the selection process more robust.

In detail, the exponential mechanism aims to select a high-scoring candidate. Given a dataset $x$, a real number $\lambda > 0$, a candidate $r \in \mathcal{R}$ is selected with a probability proportional to $e^{-u(x,r)/\lambda}$, where $u(\cdot, \cdot)$ is the utility score associated with dataset and candidate. We formalize this statement using the following lemma.

**Lemma C.1** (Utility of the Exponential Mechanism (McSherry & Talwar, 2007; Dwork et al., 2014))**.** *Let $\lambda > 0$, we use the exponential mechanism $M(\cdot)$ to select candidate $r^* \in \mathcal{R}$ for a given dataset $x$. Then for any $t > 0$ we have*

$$\Pr[u(x, r^*) \geq \mathrm{OPT} + \lambda \cdot (\ln |\mathcal{R}| + t)] \leq e^{-t},$$

*where* $\mathrm{OPT} = \min_{r \in \mathcal{R}} u(x, r)$.

### C.2. Hierarchically Well Separated Tree and CLNSS Algorithm.

We will next explain how to construct a 2-RHST. First, let us examine the properties of a 2-RHST.

**Construction of** 2-**RHST.** We now describe the construction of a 2-RHST, which involves embedding the dataset into a restricted hierarchical structure based on a quadtree. Let $D_{\max}$ and $D_{\min}$ represent the maximum and minimum Euclidean distances, respectively, between any two points in the set $P$. Let $\Lambda = \frac{D_{\max}}{D_{\min}}$. Starting with a $d$-dimensional cube of side length $\Lambda$, the space is recursively divided into $2^d$ smaller subcubes. Each subcube, with side length halved at each level, connects to its parent with an edge length proportional to $\Lambda \sqrt{d}/2^{i+1}$, where $i$ denotes the level. This process continues until the side length is reduced to 1, at which point each subcube contains exactly one point. The construction of the 2-RHST tree, denoted CONSTRUCT2RHST($P$), is presented in Algorithm 4.

The CLNSS algorithm starts by applying a random shift to the dataset $P$, where the shift is uniformly sampled from $[0, \Lambda]^d$. Then it invokes the above CONSTRUCT2RHST on the shifted points to obtain a 2-RHST $T$ and then invokes Algorithm 1 on $T$ to find the nested sequence of cluster sets. The Algorithm 1 uses a greedy method to find centers and the corresponding clustering. The final algorithm is described in Algorithm 5.

---

**Algorithm 4** CONSTRUCT2RHST($P$)  $\triangleright$Construction of 2-RHST

---

**Input:** Set of points $P$

6 **Initialize:** Construct a $d$-dimensional cube with side length $\Lambda$, where $\Lambda = \frac{D_{\max}}{D_{\min}}$, and the cube contains the entire point set $P \subseteq \{0, \ldots, \Lambda\}^d$.

7 **for** $i = 1, \ldots, \log \Lambda$ **do**

8     Recursively divide each cube along the midpoint of each dimension, generating $2^d$ subcubes per division.

9     Update the side length of each subcube to $\frac{\Lambda}{2^i}$ if the subcube contains points from the point set $P$; otherwise, delete the subcube. The remaining subcubes that contain points are treated as child nodes of their parent node.

10     Set the weight between each newly generated node and its parent node to $\Lambda \cdot \frac{\sqrt{d}}{2^i}$.

**Output:** 2-RHST tree

---

**Algorithm 5** THE CLNSS ALGORITHM

---

**Input:** Set of points $P$, distance ratio $\Lambda$

1 Apply a random shift to each point in $P$, where the shift is uniformly drawn from $[0, \Lambda]^d$.

2 Run Algorithm 4 on the shifted points to build a 2-RHST tree $T$.

3 Invoke Algorithm 1 on $T$ to produce a nested sequence of cluster sets.

**Output:** A nested sequence of cluster sets

---

# D. Deferred Lemmas and Proofs from Section 3

## D.1. Analysis of Algorithm 2: Proof of Theorem 3.1

### D.1.1. PROOF OF THEOREM 3.1: SENSITIVITY

**Lemma D.1** (From (Kumabe & Yoshida, 2022))**.** *Let $D(\cdot, \cdot)$ denote either the earth mover's distance or the total variation distance. Let* ALG *be a randomized algorithm. Suppose there is a parameter $\lambda$ (resp., $\lambda_i$) used in* ALG *or the instance $P$ (resp., $P_i$), sampled from the uniform distribution over $[B, (1 + \alpha)B]$ (resp., $[B_i, (1 + \alpha)B_i]$). Let $M$ be an upper bound of $D(\mathrm{ALG}(P), \mathrm{ALG}(P_i))$. Then for any $j > 0$, we have*

$$\frac{1}{n} \sum_{i=1}^{n} D(\mathrm{ALG}(P), \mathrm{ALG}(P^{(i)}))$$

$$\leq \frac{1}{\alpha B n} \int_{B}^{(1+\alpha)B} \left( \sum_{i=1}^{n} D\left( \mathrm{ALG}(P)\big|_{\lambda=\hat{\lambda}}, \mathrm{ALG}(P^{(i)})\big|_{\lambda^i=\hat{\lambda}} \right) \right) d\hat{\lambda} + \frac{M}{n} \cdot \frac{1 + \alpha}{\alpha} \cdot \sum_{i=1}^{n} \left| 1 - \frac{B^{(i)}}{B} \right|.$$

Lemma D.1 can be used to bound the average sensitivity when the parameter $\varepsilon$ required by the algorithm corresponds to two similar distributions.

*Proof of Lemma 3.6.* **Case 1.** If

$$\sum_{p \in P} \exp\left( \frac{-T(P, p)}{\lambda} \right) \geq \sum_{p \in P^{(i)}} \exp\left( \frac{-T^{(i)}(P^{(i)}, p)}{\lambda} \right),$$

then we can get

$$(II) = \sum_{i=1}^{n} \sum_{x \in P^{(i)}} \left( \frac{\exp\left( \frac{-T^{(i)}(P^{(i)}, x)}{\lambda} \right)}{\sum\limits_{p \in P^{(i)}} \exp\left( \frac{-T^{(i)}(P^{(i)}, p)}{\lambda} \right)} - \frac{\exp\left( \frac{-T^{(i)}(P^{(i)}, x)}{\lambda} \right)}{\sum\limits_{p \in P} \exp\left( \frac{-T^{(i)}(P, p)}{\lambda} \right)} \right)$$

$$\leq \sum_{i=1}^{n} \sum_{x \in P^{(i)}} \left( \frac{\exp\left( \frac{-T^{(i)}(P^{(i)}, x)}{\lambda} \right)}{\sum\limits_{p \in P^{(i)}} \exp\left( \frac{-T^{(i)}(P^{(i)}, p)}{\lambda} \right)} - \frac{\exp\left( \frac{-T^{(i)}(P^{(i)}, x)}{\lambda} \right)}{\sum\limits_{p \in P} \exp\left( \frac{-T^{(i)}(P^{(i)}, p)}{\lambda} \right)} \right)$$

$$= \sum_{i=1}^{n} \sum_{x \in P^{(i)}} \frac{\exp\left(\frac{-T^{(i)}(P^{(i)},x)}{\lambda}\right) \cdot \exp\left(\frac{-T^{(i)}(P^{(i)},p^i)}{\lambda}\right)}{\sum_{p \in P^{(i)}} \exp\left(\frac{-T^{(i)}(P^{(i)},p)}{\lambda}\right)} \cdot \frac{1}{\sum_{p \in P} \exp\left(\frac{-T^{(i)}(P^{(i)},p)}{\lambda}\right)}$$

$$= O(1).$$

**Case 2.** If

$$\sum_{p \in P} \exp\left(\frac{-T(P,p)}{\lambda}\right) < \sum_{p \in P^{(i)}} \exp\left(\frac{-T^{(i)}(P^{(i)},p)}{\lambda}\right),$$

then we can get

$$(II) = \sum_{i=1}^{n} \sum_{x \in P^{(i)}} \frac{\exp\left(\frac{-T^{(i)}(P^{(i)},x)}{\lambda}\right)}{\sum_{p \in P^{(i)}} \exp\left(\frac{-T^{(i)}(P^{(i)},p)}{\lambda}\right)} \cdot \left(\frac{\sum_{p \in P^{(i)}} \exp\left(\frac{-T^{(i)}(P^{(i)},p)}{\lambda}\right)}{\sum_{p \in P} \exp\left(\frac{-T^{(i)}(P,p)}{\lambda}\right)} - 1\right)$$

$$\leq O\left(\frac{\ln n}{\varepsilon}\right).$$

The final inequality follows from Lemma 3.5 and the fact that

$$\sum_{a \in A, b \in B} |a - b| \geq \sum_{a \in A, b \in B} |A - B|.$$

By combining the two cases, we can conclude that

$$(II) \leq O\left(\frac{\ln n}{\varepsilon}\right).$$

$\square$

*Proof of Lemma 3.7.* We first establish an inequality.

$$\frac{2}{n} \cdot \sum_{i=1}^{n} \left| 1 - \frac{B^{(i)}}{B} \right|$$

$$= \frac{2}{n} \sum_{i=1}^{n} \frac{6 \ln n}{\varepsilon \cdot T(P,\bar{x}_t)} \cdot \left| \frac{\varepsilon \cdot T(P,\bar{x}_t)}{6 \ln n} - \frac{\varepsilon \cdot T^{(i)}\left(P^{(i)},\bar{x}_t^{(i)}\right)}{6 \ln(n-1)} \right|$$

$$\leq \frac{2}{n} \sum_{i=1}^{n} \frac{\ln n}{\varepsilon \cdot T(P,\bar{x}_t)} \max\left\{ 0, \frac{\varepsilon \cdot T(P,\bar{x}_t)}{\ln n} - \frac{\varepsilon \cdot T^{(i)}\left(P^{(i)},\bar{x}_t^{(i)}\right)}{\ln(n-1)} \right\}$$

$$+ \frac{2}{n} \sum_{i=1}^{n} \frac{\ln n}{\varepsilon \cdot T(P,\bar{x}_t)} \cdot \max\left\{ 0, \frac{\varepsilon \cdot T^{(i)}\left(P^{(i)},\bar{x}_t^{(i)}\right)}{\ln(n-1)} - \frac{\varepsilon \cdot T(P,\bar{x}_t)}{\ln n} \right\}$$

For the first inequality we have

$$\frac{2}{n} \sum_{i=1}^{n} \frac{\ln n}{\varepsilon \cdot T(P,\bar{x}_t)} \cdot \max\left\{ 0, \frac{\varepsilon \cdot T(P,\bar{x}_t)}{\ln n} - \frac{\varepsilon \cdot T^{(i)}\left(P^{(i)},\bar{x}_t^{(i)}\right)}{\ln(n-1)} \right\}$$

$$\leq \frac{2}{n} \sum_{i=1}^{n} \frac{\ln n}{\varepsilon \cdot T(P,\bar{x}_t)} \cdot \left( \frac{\varepsilon \cdot T(P,\bar{x}_t)}{\ln(n-1)} - \frac{\varepsilon \cdot T^{(i)}\left(P^{(i)},\bar{x}_t^{(i)}\right)}{\ln(n-1)} \right)$$

$$\leq \frac{4}{n \cdot T(P, \bar{x}_t)} \cdot \left( \sum_{i=1}^{n} T(P, \bar{x}_t) - T^{(i)}(P^{(i)}, \bar{x}_t^{(i)}) \right)$$

$$\leq \frac{4}{n \cdot T(P, \bar{x}_t)} \cdot \sum_{i=1}^{n} \left( T(P, \bar{x}_t) - T(P, \bar{x}_t^{(i)}) + T(x^{(i)}, \bar{x}_t^{(i)}) \right)$$

$$\leq \frac{4}{n \cdot T(P, \bar{x}_t)} \cdot \sum_{i=1}^{n} T(x^{(i)}, \bar{x}_t) + T(\bar{x}_t^{(i)}, \bar{x}_t)$$

$$\leq O(\frac{1}{n})$$

For the second inequality we have

$$\frac{2}{n} \sum_{i=1}^{n} \frac{\ln n}{\varepsilon \cdot T(P, \bar{x}_t)} \cdot \max \left\{ 0, \frac{\varepsilon \cdot T^{(i)}\left(P^{(i)}, \bar{x}_t^{(i)}\right)}{\ln(n-1)} - \frac{\varepsilon \cdot T(P, \bar{x}_t)}{\ln n} \right\}$$

$$\leq \frac{2}{n} \sum_{i=1}^{n} \frac{\ln n}{\varepsilon \cdot T(P, \bar{x}_t)} \cdot \max \left\{ 0, \frac{\varepsilon \cdot T^{(i)}\left(P^{(i)}, \bar{x}_t^{(i)}\right)}{\ln(n-1)} - \frac{\varepsilon \cdot T^{(i)}\left(P^{(i)}, \bar{x}_t^{(i)}\right)}{\ln n} \right\}$$

$$\leq \frac{2}{n} \sum_{i=1}^{n} \frac{\ln n}{\varepsilon \cdot T(P, \bar{x}_t)} \cdot \left( \frac{\varepsilon \cdot T^{(i)}\left(P^{(i)}, \bar{x}_t^{(i)}\right)}{\ln(n-1)} - \frac{\varepsilon \cdot T^{(i)}\left(P^{(i)}, \bar{x}_t^{(i)}\right)}{\ln n} \right)$$

$$\leq \frac{2}{n} \sum_{i=1}^{n} \ln n \cdot \left( \frac{1}{\ln(n-1)} - \frac{1}{\ln n} \right)$$

$$= \frac{2}{n} \cdot \left( \frac{n \ln n}{\ln(n-1)} - n \right) = \frac{2}{n} \cdot O(1) = O(\frac{1}{n})$$

Here, $\frac{n \ln n}{\ln(n-1)} - n$ is a decreasing function, and its value is less than 1 when $n > 10$. Combining the two inequalities above, we obtain

$$\frac{2}{n} \sum_{i=1}^{n} \left| 1 - \frac{B^{(i)}}{B} \right| \leq O(\frac{1}{n}).$$

$\square$

### D.1.2. PROOF OF THEOREM 3.1: TIME COMPLEXITY

The time complexity of the algorithm can be analyzed in two parts.

First, we consider the cost of constructing the 2-RHST. In the 2-RHST tree partitioning, each point belongs to exactly one node at each layer, and each layer partitions each node precisely once. Consequently, each point is divided at most $O(d\Lambda)$ times. Given $n$ points, the time complexity for this partitioning process is $O(nd\Lambda)$.

Next, we analyze the time complexity of hierarchical clustering. At each layer, selecting the center using the exponential mechanism incurs a time complexity of $O(n^2)$. Since this process is repeated $n$ times, the overall clustering complexity is $O(n^3)$.

Combining these two components, the total time complexity of the algorithm is:

$$O(dn \log \Lambda + n^3).$$

It is important to note that the time and space complexity of Algorithm 2 remain the same as in the original CLNSS version (Algorithm 1).

## D.2. Proof of Corollary 3.2

Combining Theorem 3.1 and a technique from (Cohen-Addad et al., 2021), we are ready to give the proof of Corollary 3.2.

*Proof of Corollary 3.2.* First, note that the cost of any valid solution is bounded below by 1 and above by $d\Lambda n$. Let $k_1, k_2, \ldots, k_m$ be the sequence such that $k_i$ is the largest number of centers such that the optimal $k_i$-median cost is at least $2^i$, here $m = \log(d\Lambda n)$. Thus, according to Theorem 3.1, we have

$$\mathbb{E}\left[\max_{i \in [m]} \frac{\mathrm{COST}_T(P, S_{k_i}, k_i)}{\mathrm{OPT}(P, k_i)}\right] \leq m \cdot \max_{i \in [m]} \mathbb{E}\left[\frac{\mathrm{COST}_T(P, S_{k_i}, k_i)}{\mathrm{OPT}(P, k_i)}\right] = O(md \log \Lambda \cdot (1 + \varepsilon)^k).$$

Now, observe that for any $i$ and any $k$ with $k_i < k < k_{i-1}$, we have $\mathrm{COST}_T(P, S_k, k) = O(\mathrm{COST}_T(P, S_{k_{i-1}}, k_{i-1}))$ since $\mathrm{COST}_T(P, S_k, k)$ is non-increasing in $k$ and we have $2 \cdot \mathrm{OPT}(P, k) \leq \mathrm{OPT}(P, k_i)$ by the choice of the $k_i$. This implies that

$$\frac{\mathrm{COST}_T(P, S_k, k)}{\mathrm{OPT}(P, k)} = O\left(\frac{\mathrm{COST}_T(P, S_{k_i}, k_i)}{\mathrm{OPT}(P, k_i)}\right).$$

This completes the proof. □

## E. Proof of Lemma 4.2

Now, we present the proof of Lemma 4.2.

*Proof.* For simplicity, assume the number of clusters $k$ is 2. Consider a pair of adjacent 2-dimensional datasets $X$ and $X'$, where $X'$ is obtained by removing a point from $X$. The first three layers of their RHST trees and the subtrees containing the first two selected centers are shown in Fig. 6. In both of these RHST trees, subtrees 1 to 16 each contain $\Omega(n)$ points.

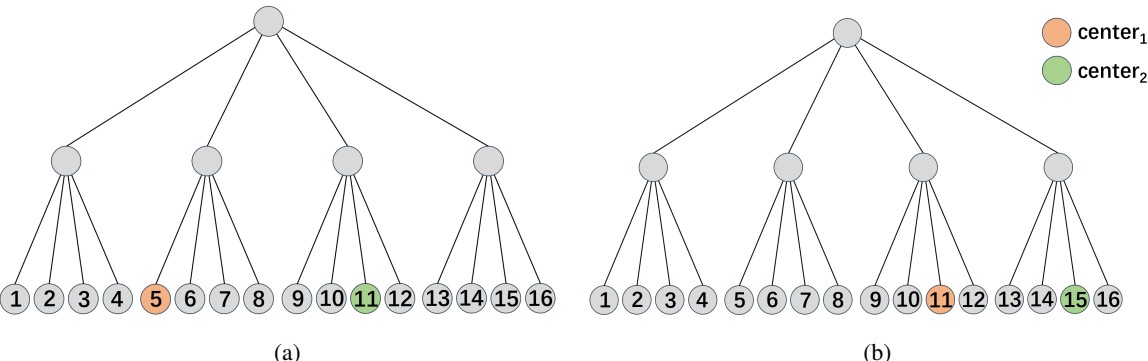

*Figure 6.* Fig. 6a is the clustering of the 2-RHST constructed from the original dataset when $k = 2$, while Fig. 6b is the clustering of the 2-RHST after deleting a single point, also with $k = 2$.

Assume that in the first RHST tree, a point chosen as the center in subtree 5 has the minimum COST, while points in subtree 11 and subtree 15 chosen as centers have the second and third smallest COST, with a difference of less than $\frac{\Lambda}{2}$ from the minimum. And, the COST of points chosen as centers from other subtrees is somewhat higher. Then, after deleting any point in subtree 5, the first two centers will change to points in subtree 11 and subtree 15, as shown in Fig. 6. Note that such examples are easy to construct. Thus the average sensitivity on the above graph is at least $\frac{1}{n} \cdot \Omega(n) \cdot \Omega(n) = \Omega(n)$. □

## F. More on Experiments

### Synthetic Datasets

We will now provide additional details on the data generation settings used in the experiment.

(1) Specific instances: we first generate synthetic data as described in Lemma 4.1 and Lemma 4.2.

In the experiments comparing our algorithm with the CLNSS algorithm and single linkage on these datasets, we set the value of $\varepsilon$ to 1, $k$ to 2, and the dataset sizes ranged from 50 to 300 in increments of 50. Specifically, we generated two types of datasets: In Fig. 2a, the data points start from the coordinate origin and are distributed along a random line with progressively increasing distance intervals, while in Fig. 2b, the data points are concentrated at a position in the Euclidean space corresponding to the third-layer nodes of the RHST tree, with settings identical to those described in the lemmas.

(2) Random datasets: In these experiments Fig. 3, $\varepsilon$ takes the values 1, 10, and 1000, with the dataset size fixed at 500 points. We conducted hierarchical clustering experiments, as shown in Fig. 3, on a regression dataset generated using scikit-learn (Pedregosa et al., 2011). The results of the experiments on synthetic datasets were averaged over 50 independent trials. Since our algorithm is stochastic, each trial was repeated 10 times to compute the average.

**Real Datasets** Table Table 1 summarizes the real datasets used in the experiments. It lists the dataset names along with their

| Dataset | n | d | clusters |
|---|---|---|---|
| Iris | 150 | 4 | 3 |
| Wine | 178 | 13 | 3 |
| Wholesale | 440 | 6 | 2 |
| Diabetes | 442 | 10 | 2 |
| WDBC | 569 | 30 | 2 |
| Digits | 1797 | 64 | 10 |
| Yeast | 2417 | 24 | 10 |

*Table 1.* Summary of the datasets used in the experiments. Here, $n$ represents the size of the dataset, $d$ denotes its original dimensionality, and clusters indicate the number of classifications in the dataset.

respective sizes ($n$), original dimensionalities ($d$), and the number of clusters. For high-dimensional datasets (e.g., Digits and Yeast), dimensionality reduction was performed prior to analysis.

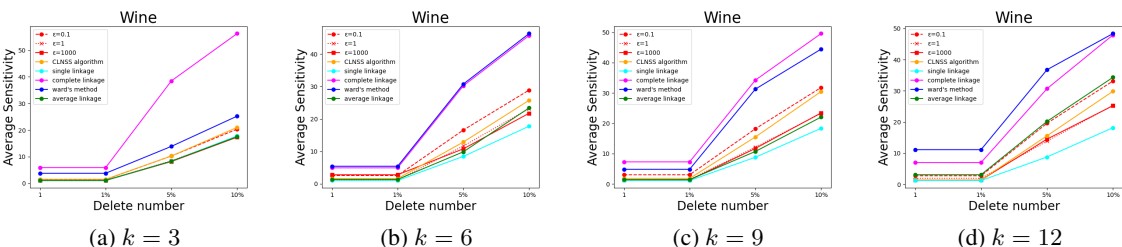

(a) $k = 3$     (b) $k = 6$     (c) $k = 9$     (d) $k = 12$

*Figure 7.* Results for different values of $k$ on the Wine dataset. Our algorithm exhibits lower average sensitivity across all values of $k$ in the graph, whereas other algorithms (except for single linkage) perform better only for specific values of $k$.

We also evaluated the performance of the algorithm on the same dataset for different values of $k$ (see Fig. 7). Fig. 7 shows that the average sensitivity of our algorithm for certain values of $k$ is quite small. Additionally, we can see complete linkage, Ward's method, and the CLNSS algorithm exhibit low average sensitivity only for specific values of $k$.

The appendix also includes the average sensitivity for certain datasets with respect to a given value of $k$ (Fig. 8). Since the Yeast dataset is larger, we chose a higher value of $k$ to evaluate the average sensitivity. Additionally, we provide the $k$-median cost for a continuous range of $k$ values across the three datasets (Fig. 9).

## G. Low Average Sensitivity of Single Linkage on Well-Clusterable Datasets

We introduce the following definition of well-clusterable datasets.

**Definition G.1** (Well-clusterable datasets)**.** *Let $m \leq n$ be an integer. Let $P$ be a set of $n$ points, and let $\{C_1, \ldots, C_m\}$ be a partition of $P$ into $m$ clusters. Let $d_i$ represent the maximum weight of edges in the minimum spanning tree (MST) of cluster $C_i$, and let $d_{i,j} = \min_{x \in C_i, y \in C_j} DIST(x, y)$ denote the minimum distance between clusters $C_i$ and $C_j$.*

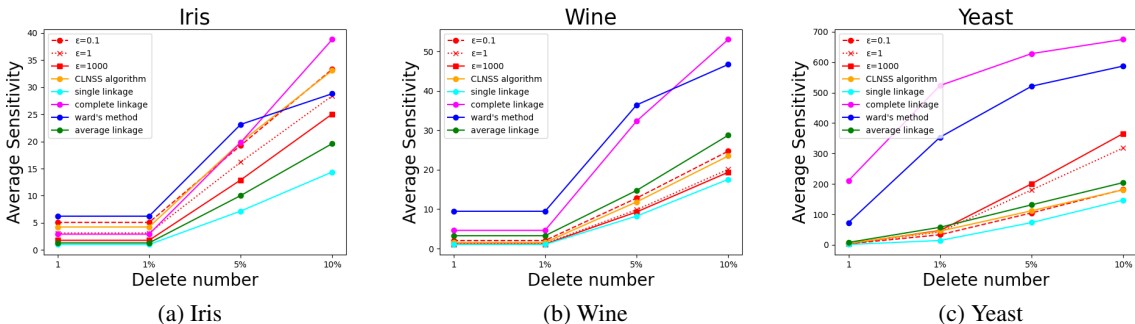

(a) Iris  (b) Wine  (c) Yeast

*Figure 8.* Results on real-world data sets. The $x$-axis represents the deleted points number and the $y$-axis represents the corresponding average sensitivity. Here, $k$ is fixed: Iris $k = 4$, Wine $k = 4$, Yeast $k = 12$.

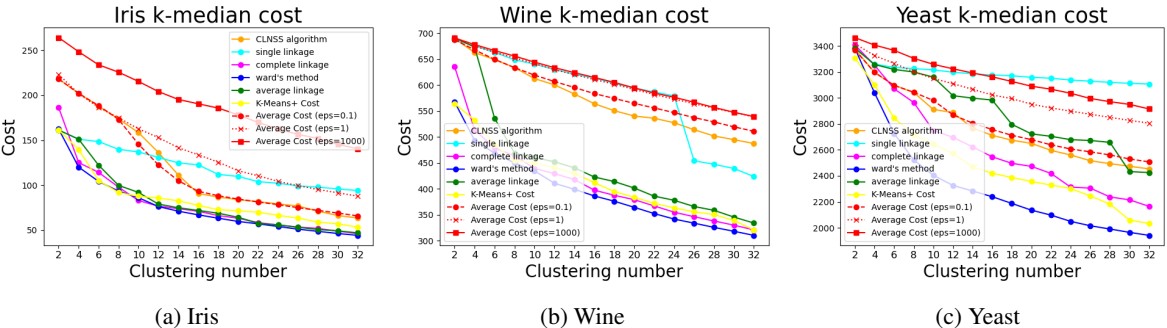

(a) Iris  (b) Wine  (c) Yeast

*Figure 9.* Results on real-world data sets. The $x$-axis represents the centers, and the $y$-axis represents the $k$-median cost corresponding to the $k$ centers. On the Iris dataset, our algorithm gradually outperforms the single linkage algorithm as $k$ increases and performs similarly to other algorithms. On the Wine dataset, our algorithm closely matches the performance of CLNSS and single linkage. On the WDBC dataset, only the single linkage algorithm exhibits poor performance.

*We say $P$ is a well-clusterable dataset if the distance between any two clusters $C_i$ and $C_j$ satisfies:*

$$d_{i,j} > 2\max(d_i, d_j), \quad \forall i \neq j.$$

We provide the following guarantee for single linkage clustering on well-clusterable datasets: the single linkage clustering exhibits small average sensitivity for $k$-clustering if $m$ is small and $k \leq m$.

**Lemma G.2.** *Let $P$ be a well-clusterable dataset. Then for $1 \leq k \leq m$, the sensitivity of the $k$-clustering result of single linkage is bounded by $2(m - k) + 1$, which is at most $2m$, where $m$ is as specified in Definition G.1.*

*Proof.* Let $P' = \{C_1, \ldots, C_i - x, \ldots, C_m\}$ be the data obtained by deleting a vertex $x$ from $C_i$. We need to measure the difference in the single linkage $k$-clustering results between $P$ and $P'$.

First, consider the case when $k = m$. Note that the process of single linkage is essentially the same as Kruskal's algorithm for constructing the MST, and since $d_{i,j} > 2\max(d_i, d_j)$, no clusters $C_i$ and $C_j$ (with $i \neq j$) will be merged before $C_1, \ldots, C_m$ are clustered separately. Therefore, the single linkage $k$-clustering result of $P$ is $\{C_1, \ldots, C_m\}$. For $P'$, if $x$ is a leaf in the MST of $C_i$, deleting it will not affect $d_i$. If $x$ is not a leaf in the MST of $C_i$, deleting it will split the original MST into several trees. We then consider connecting these trees in an arbitrary way to form a new spanning tree for $C_i$. By the triangle inequality, the newly added edges will not exceed $2d_i$ and thus will be less than $d_{i,j}$ for any $j \neq i$. Therefore, the clustering result for $P'$ is $\{C_1, \ldots, C_i - x, \ldots, C_m\}$, and the sensitivity of the $k$-clustering is 1.

For $k = m - 1, \ldots, 1$, this is equivalent to performing $m - k$ additional single linkage merges starting from the $k$-clustering result. Since in each single linkage merge, the inter-cluster distance is affected by only two vertices, and deleting other vertices will not increase the inter-cluster distance between clusters, the single linkage merge will not be affected as long as

| Dataset | $\varepsilon$ | min_samples |
|---------|----|-------------|
| Wholesale | 0.1 | 3 |
| Diabetes | 0.6 | 3 |
| Digits | 0.5 | 3 |
| Yeast | 0.5 | 3 |

*Table 2.* Parameter settings of the DBSCAN algorithm for various datasets, including details of the $\varepsilon$ and min_samples parameters for each dataset. The parameter $\varepsilon$ defines the neighborhood range (neighborhood radius) and determines whether points are considered neighbors, while min_samples specifies the minimum number of neighbors required for a point to be classified as a core point, thus controlling the density of the cluster.

the deleted point $x$ is not one of these two points. Thus, the average sensitivity can be bounded as follows:

$$1 \times \frac{n - 2(m-k)}{n} + n \times \frac{2(m-k)}{n} \leq 2(m-k) + 1,$$

where the first term $1 \times \frac{n-2(m-k)}{n}$ corresponds to deleting vertices that do not affect the single linkage merge, and the second term $n \times \frac{2(m-k)}{n}$ accounts for deleting vertices that do affect the merge. □

**Many real datasets closely resemble well-clusterable ones**    To further validate the clustering structure, we applied the DBSCAN algorithm[3], a classic density-based clustering method. We chose DBSCAN over the original labels because the dataset contains outliers, which cause the clusters defined by the labels to be poorly connected. As a density-based algorithm, DBSCAN is better suited to handle such cases, as it preserves the inherent structure of the clusters.

To use DBSCAN, two parameters must be set: $\varepsilon$ and min_samples. $\varepsilon$ defines the neighborhood radius and determines whether points are considered neighbors, while min_samples specifies the minimum number of neighbors required for a point to be classified as a core point, thereby controlling the density of the clusters. DBSCAN begins by examining each point in the dataset. If a point has enough neighboring points within its $\varepsilon$-radius (i.e., at least min_samples neighbors), it is designated as a core point, and a cluster is formed. Points that are within the $\varepsilon$-radius of a core point but do not themselves have enough neighbors are classified as border points.

We use the output clustering of the DBSCAN algorithm to calculate the max-intra distance $d_i$ and the inter-cluster distance $d_{i,j}$, as defined in Definition G.1.

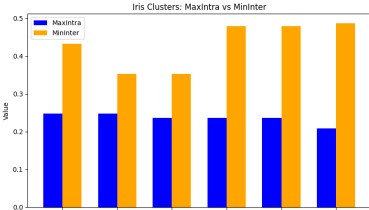

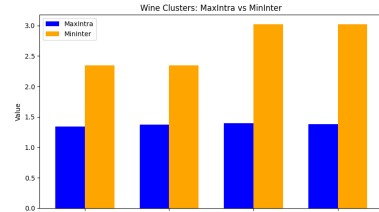

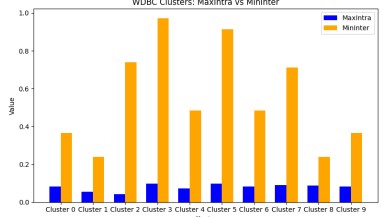

(a) In the Iris dataset, the DBSCAN parameters are set to $\varepsilon = 0.25$ and min_samples $= 3$.

(b) In the Wine dataset, the DBSCAN parameters are set to $\varepsilon = 1.5$ and min_samples $= 3$.

(c) In the WDBC dataset, the DBSCAN parameters are set to $\varepsilon = 0.1$ and min_samples $= 3$.

*Figure 10.* The $x$-axis represents the number of clusters, while the $y$-axis corresponds to the Euclidean distance. MaxIntra denotes $d_i$, while MinInter represents the minimum inter-cluster distance among $d_{i,j}$ values adjacent to $d_i$. Experiments show that these dataset exhibit a strong clustering structure, i.e., there is large gap between MaxIntra and MinInter, which explains why single linkage achieves lower average sensitivity on such datasets.

The experimental results in Fig. 10 confirm our hypothesis that these real-world datasets (Iris, Wine and WDBC) exhibit a strong cluster structure, i.e. closely resemble well-clusterable datasets. Thus, single linkage exhibits lower sensitivity. Other

---

[3]Note that we do not use DBSCAN for preprocessing the data; it is only used to verify whether the real-world dataset is well-clusterable. Other clustering methods can also be employed to assess the clusterability property. For the evaluation of our algorithms, we apply our clustering method and other baselines directly to the original dataset.

datasets exhibit similar properties, however, due to variations in the number of well-defined clusters across datasets and the difficulty of visualizing large datasets in graphs, we provide the settings Table 2 for reproducibility, allowing readers to verify the results themselves.

