# OpenReview forum: "Average Sensitivity of Hierarchical $k$-Median Clustering"
_ICML.cc/2025/Conference — ICML 2025 poster_

### Official Review · Reviewer_7WCS · 2025-03-12

**Overall Recommendation:** 3

**Summary:**

This paper studies the hierarchical $k$-median problem in the setting of average sensitivity, which is a measure of how much an algorithm's output changes when the dataset undergoes small perturbations. The paper's first contribution is algorithmic, and it proposes an efficient algorithm for the hierarchical $k$-median problem, with rigorous theoretical guarantees with respect to the average sensitivity and the expected cost of the returned solution. At a technical level, their algorithm combines the CLNSS algorithm [Cohen-Addad et al., 2021] with an exponential mechanism. The paper's second contribution is 2 results on the worst-case average sensitivity of (i) single linkage and (ii) the CLNSS algorithm. The paper's third contribution is an experimental evaluation of their proposed algorithm, which they compare against a number of different HC algorithms on a variety of synthetic instances and datasets from the Scikit-learn repo and UCI ML repo. As a final small result, they show that if the data points are well-clustered, then single linkage has provably much lower average sensitivity.

**Claims And Evidence:**

The main theoretical results/claims are supported by proofs. I checked these proofs, and did not find any errors.

The experimental section does contain some claims which are not supported by the presented evidence. In particular, in the second to last paragraph it is stated that "(...) Fig. 5 shows that our algorithm ($k=4$) outperformls traditional linkage-based algorithms and the CLNSS algorithm (Recall that it achives lower clustering costs than average linkage, single linkage, and the CLNSS algorithm, as seen in Figs. 4 and 9)." However, this claim seems overly broad. For example, on the iris dataset, average linkage outperforms the proposed method both with respect to the average sensitivity and with respect to the clustering cost. On the wine dataset, average linkage achieves much better cost.

Further on in the experimental section, it is mentioned that "our algorithm consistently maintains lower average sensitivity across different values of $k$, whereas other algorithms (...) perform well only for certain $k$ values.". Again, this claim is slightly exaggerated, as there are a number of algorithms (e.g., average linkage) where the average sensitivity is also consistent for different values of $k$.

**Essential References Not Discussed:**

N/A

**Experimental Designs Or Analyses:**

The one issue I found are the experiments corresponding to Figure 2. Why are single linkage and CLNSS the only algorithms that are compared on the synthetic instances? Furthermore, Fig 2a and Fig 2b evaluate different synthetic instances. I would expect all methods to be evaluated on the synthetic data, similar to what is done in the experiments on real-world data.

**Methods And Evaluation Criteria:**

Yes, the experimental setup is sufficiently good in terms of what benchmark datasets are used, and what algorithms are compared against.

**Other Comments Or Suggestions:**

1. line 067, right column: "agglomeritive" --> should be "agglomeration".

2. line 423, right column: "our our" --> should be "our".

3. Equations that exceed the column width, e.g., in Corollary 3.2, on lines 235--242, left column, on lines 240 -- 245 in the right column, and on lines 362 in the left column.

3. $P^{(i)}$ is defined again in Section 3, whereas it was already introduced in the preliminary section.

**Other Strengths And Weaknesses:**

S1) The proposed algorithm has good theoretical guarantees. In particular, the average sensitivity bound is significantly better than some of the worst case bound on popular algorithms (single linkage for example). To the best of my knowledge, this is the first such result, making it interesting.

S2) I think the results on the lower bound (and upper bound for well-clustered data) are quite nice and insightful. This sheds light on when these types of algorithms should or shouldn't be used if a user is interested in average sensitivity.

W1) The main weakness in my opinion is the lack of novelty in the result. Currently, the result seems like a fairly straightforward application of the exponential mechanism in the CLNSS algorithm - not much other technical novelty is needed.

W2) The write-up could be improved. For example, the main text refers to the appendix a lot, requiring the reader to go back and forth to check the correctness of the result. As a suggested improvement, I would instead move most of the theorem/lemma statements that are needed into the main text, and instead move all the proof environments to the appendix.

**Questions For Authors:**

1.) In Theorem 3.1, is the dependence on $k$ necessariy in the approximation factor and average sensitivity bounds? For low $k$, this result is quite strong, however, as $k$ approaches $O(n)$ the guarantees become much worse.

2.) Could you please elaborate on the main technical novelty of the result? As mentioned above, it currently seems that the main result seems like an application of the exponential mechanism to CLNSS. Are there any significant difficulties that need to be overcome? Or can the result be applied directly?

**Relation To Broader Scientific Literature:**

The paper makes clear contributions to the broader literature on clustering robustness and algorithmic stability, also studied in works by Peng and Yoshida (2020), Varma & Yoshida (2021), and Yoshida & Ito (2022) who have studied the average sensitivity for various clustering and graph-theoretic problems. The authors generalise and strengthen the previous CLNSS algorithm by integrating the exponential mechanism [McSherry & Talwar, 2007] for differential privacy to systematically control the average sensitivity in hierarchical clustering. Therefore one could see this work as a bridge between literature on hierarchical clustering (e.g., Dasgupta, 2016; Moseley & Wang, 2023) and robustness/differential privacy (Imola et al., 2023; Cohen-Addad et al., 2021, 2022).

**Theoretical Claims:**

Yes I did, no issues found.

---

> ### Author Rebuttal · Authors · 2025-03-30
>
> Thank you for your suggestion. We fixed the typos in the updated manuscript. We will address your concerns as follows:
>
> **Claims And Evidence:**
>
> **C1: The experimental section does contain some claims which are not supported by the presented evidence.**
>
> We appreciate your careful review and will revise the experimental section accordingly.
>
> **Experimental Designs Or Analyses:**
>
> **E1: The one issue I found are the experiments corresponding to Figure 2. Why are single linkage and CLNSS the only algorithms that are compared on the synthetic instances? Furthermore, Fig 2a and Fig 2b evaluate different synthetic instances. I would expect all methods to be evaluated on the synthetic data, similar to what is done in the experiments on real-world data.**
>
> We sincerely appreciate your feedback and will carefully consider your suggestions. The two datasets in Fig. 2 were specifically chosen as challenging instances based on Lemma 4.1 and Lemma 4.2 to evaluate the theoretical lower bounds of the single linkage and CLNSS algorithms. We initially thought it might not be particularly insightful to include other methods in this context.
>
> However, we will evaluate the performance of other methods on these two datasets in the revised version to provide a more comprehensive comparison.
>
> **Weaknesses:**
>
> **W1: The main weakness in my opinion is the lack of novelty in the result. Currently, the result seems like a fairly straightforward application of the exponential mechanism in the CLNSS algorithm - not much other technical novelty is needed.**
>
> We noticed that this weakness is similar to Q2. We have provided a detailed response in Q2.
>
> **W2: The write-up could be improved. For example, the main text refers to the appendix a lot, requiring the reader to go back and forth to check the correctness of the result. As a suggested improvement, I would instead move most of the theorem/lemma statements that are needed into the main text, and instead move all the proof environments to the appendix.**
>
> Thank you for your feedback. We will move most of the theorem and lemma statements into the main text and shift the proofs to the appendix as suggested.
>
>
> **Q1: In Theorem 3.1, is the dependence on $k$ necessariy in the approximation factor and average sensitivity bounds? For low $k$, this result is quite strong, however, as $k$ approaches $O(n)$ the guarantees become much worse.**
>
> Thank you for raising this important point. We agree with your observation that as $k$ becomes large, our guarantees do indeed weaken. While it is unclear whether the dependency on $k$ in the utility is strictly necessary, we believe that the dependency on average sensitivity is essential. Since hierarchical clustering is constructed from top to bottom, it is intuitive that the number of misclassified points will accumulate, leading to a dependency on the layer $k$ in the sensitivity bound.
>
> For the utility part, we feel that our analysis might be somewhat conservative. The accuracy loss occurs by a $(1 + \epsilon)$ factor at each layer in our induction approach, leading to a  $(1 + \epsilon)^k$ factor. There may be an opportunity to refine the analysis to mitigate the exponential dependence on $k$, which we see as an interesting future direction.
>
> Finally, we note that in our experiments (Fig. 4 & Fig. 9), the approximation ratio does not increase exponentially, suggesting that there is potential for further improvement in our approximation ratio.
>
> **Q2: Could you please elaborate on the main technical novelty of the result? As mentioned above, it currently seems that the main result seems like an application of the exponential mechanism to CLNSS. Are there any significant difficulties that need to be overcome? Or can the result be applied directly?**
>
> Thank you for your thoughtful comments. Indeed, we have applied the exponential mechanism based on the CLNSS algorithm. While the use of the exponential mechanism to stabilize algorithms is not new (it has appeared in the differential privacy literature, for example), its recursive application to derive sensitivity bounds for hierarchical clustering is novel.
>
> One main difficulty is in bounding the aggregated error after applying the exponential mechanism at each level. For instance, at each $k$, there is an optimal $k$-median solution $\textup{OPT}(k)$, but the algorithm can only provide a local optimum. Specifically, given the current  $k-1$ clustering $\mathrm{Alg}(k-1)$, the algorithm selects a new center, resulting in a $k$-clustering $\mathrm{Alg}(k)$. The exponential mechanism can only provide a bound on the error between $\mathrm{Alg}(k)$  and $\mathrm{OPT}_k'$, the optimal $k$-clustering given $\mathrm{Alg}(k-1)$. Relating this error bound to the error bound between $\mathrm{Alg}(k)$ and $\mathrm{OPT}(k)$ is a key challenge.
>
> We address this challenge by carefully leveraging the properties of the $2$-RHST and using an inductive approach. This forms one of the novel aspects of our analysis.

---

> > ### Comment · Reviewer_7WCS · 2025-04-02
> >
> > Thank you for your clarifications - if the promised edits to the papers are implemented then I would be happy for the paper to be accepted.

---

### Official Review · Reviewer_F12p · 2025-03-13

**Overall Recommendation:** 3

**Summary:**

This study provides an innovative solution that enhances both the interpretability and robustness of hierarchical clustering techniques. The study shows that classical methods have high sensitivity on specific datasets, and validates the robustness of the new algorithm through experiments.

**Claims And Evidence:**

Yes

**Essential References Not Discussed:**

No

**Experimental Designs Or Analyses:**

Yes

**Methods And Evaluation Criteria:**

Yes

**Other Comments Or Suggestions:**

No

**Other Strengths And Weaknesses:**

Strengths:
- Theoretical Rigor: Formal proofs for sensitivity bounds and approximation ratios.
- Comprehensive Experiments: Validation on synthetic and real datasets aligns theory with practice.
- Innovative Comparison: Systematic analysis of classical methods’ limitations (e.g., Single Linkage).

Weaknesses:
- Computational Complexity: $O(n^3 )$ time complexity limits scalability to large-scale datasets.
- Experimental Bias: Reliance on DBSCAN for defining “well-clusterable” real data may introduce preprocessing bias.

**Questions For Authors:**

- The algorithm has $(O(n^3)$ time complexity. Have you considered sampling-based optimizations for large-scale data?
- How to design adaptive strategies for $\varepsilon$ (e.g., dynamically adjusting based on data distribution) instead of manual tuning?
- Can this approach extend to non-Euclidean metric spaces (e.g., graph data)?

**Relation To Broader Scientific Literature:**

This paper specifies that several agglomeritive clustering, including single linkage clustering and a variant of the CLNSS algorithm are unstable in the face of data perturbations, and points out that some other methods that consider stability are based on the identification and processing of anomalies. The authors argue that even this consideration is somewhat one-sided and propose a method based on the exponential mechanism that has a better average sensitivity.

**Theoretical Claims:**

Yes

---

> ### Author Rebuttal · Authors · 2025-03-30
>
> Thank you for your review. We will address your concerns as follows:
>
> **Weaknesses:**
>
> **W1: Computational Complexity: $O(n^3)$ time complexity limits scalability to large-scale datasets.**
>
> We noticed that this weakness is similar to Q1. Please refer to Q1 for further details.
>
> **W2: Experimental Bias: Reliance on DBSCAN for defining “well-clusterable” real data may introduce preprocessing bias.**
>
>  Please note that we do not use DBSCAN for preprocessing the data; it is only used to verify whether the real-world dataset is well-clusterable. Other clustering methods can also be employed to assess the clusterability property.
>
> For the evaluation of our algorithms, we apply our clustering method and other baselines directly to the original dataset.
>
> **Q1: The algorithm has $O(n)^3$ time complexity. Have you considered sampling-based optimizations for large-scale data?**
>
> One of the primary reasons for the  $O(n^3)$  time complexity is that the algorithm involves $n$ iterations, and in each iteration, we sequentially compute the $k$-median cost for $n$ possibilities and sample from the distribution underlying the exponential mechanism. It is challenging to optimize this process through sampling. While importance sampling could potentially help identify significant points, we require the information for all points to construct the hierarchical tree. Other sampling approaches might help, but we are not aware of any that provide significant improvements.Instead,we focused on parallelization in our experiments to enhance the algorithm’s efficiency.
>
> **Q2: How to design adaptive strategies for $\epsilon$ (e.g., dynamically adjusting based on data distribution) instead of manual tuning?**
>
> The choice of the  $\epsilon$  parameter largely depends on the specific requirements of the problem. It should be set to balance the trade-off between the approximation ratio and average sensitivity, depending on the desired outcome. For instance, in practice, one could use a geometric search to find an appropriate $\epsilon$  that satisfies the target accuracy or achieves the desired sensitivity.
>
> **Q3: Can this approach extend to non-Euclidean metric spaces (e.g., graph data)?**
>
> Indeed, the hierarchical Euclidean  $k$ -median can be generalized to metric spaces, and there are tree embedding approaches for general metric spaces. Thus, we believe it is possible to extend our approach to non-Euclidean metric spaces.
>
> However, for graph data, it is unclear how to effectively utilize the $2$-RHST tree or similar tree embeddings, as graph data typically do not contain explicit distance information or may not satisfy the triangle inequality.

---

> > ### Comment · Reviewer_F12p · 2025-04-04
> >
> > I appreciate the details and clarifications provided by the authors. I have no more concerns and will keep the rating

---

### Official Review · Reviewer_mMdj · 2025-03-17

**Overall Recommendation:** 3

**Summary:**

Hierarchical clustering is a widely used method for unsupervised learning with numerous applications. However, in the application of modern algorithms, the datasets studied are usually large and dynamic. If the hierarchical clustering is sensitive to small perturbations of the dataset, the usability of the algorithm will be greatly reduced.

This paper focuses on the hierarchical K-median clustering problem, which bridges hierarchical and centroid-based clustering while offering theoretical appeal, practical utility, and improved interpretability. We analyze the average sensitivity of algorithms for this problem by measuring the expected change in the output when a random data point is deleted. We propose an efficient algorithm for hierarchical
-median clustering and theoretically prove its low average sensitivity and high clustering quality. Additionally, we show that single linkage clustering and a deterministic variant of the CLNSS algorithm exhibit high average sensitivity, making them less stable. Finally, we validate the robustness and effectiveness of our algorithm through experiments.

**Claims And Evidence:**

Yes.

**Essential References Not Discussed:**

The authors have done a great job citing and discussing related papers. However, I think there should be a more extensive discussion on the algorithm's inherent connection to Differential Privacy (DP) algorithms; as the definition of perturbation by one point is exactly that of neighboring datasets in DP, and exponential mechanism is a universally applied DP mechanism. In that sense, can we reduce one of these problems to another? For example if we split the privacy budget among K layers, and apply exponential mechanism to them does it give us similar bounds?

**Experimental Designs Or Analyses:**

The experiments seem well-designed.

**Methods And Evaluation Criteria:**

Yes

**Other Comments Or Suggestions:**

See questions for authors.

**Other Strengths And Weaknesses:**

Weakness: the new algorithm design is relatively simple: to introduce exponential mechanism to the existing 2-RHST tree based clustering algorithm. This incurs an additional $(1+\epsilon)^k$ factor cost in the approximation ratio, which could be a lot if K is big. The results share a lot of similarities with differential privacy problems, hence not very surprising.

**Questions For Authors:**

1. Can the methods here be applied to "flat" K-clusterings? I suppose the cost function's definition must change then because there is no path.
2. The cost function seems insensitivity to relative differences among points. For example, if I take a cluster in the K clusters and move it farther away the constructed tree still seems to be the same (although some merges might change their orders). I wonder if you have any opinion about what this means if we use the traditional cost functions for K-median (sum of distances to closest center).
3. The paper starts with constructing the 2-RHST tree. The hierarchical clustering then builds on the tree only and ignores the original datasets. Intuitively why do we choose the 2-RHST tree? Are there other tree embeddings that can work for this problem?

**Relation To Broader Scientific Literature:**

The paper is closely related to the recent advances in hierarchical K-median, including tree-based agglomerative clustering methods and approximation to the optimal solution. The definition of robustness, measured by perturbation of one point, and the exponential mechanism closely related to the field of Differential Privacy.

**Theoretical Claims:**

I've checked the argument for the approximation bounds and it seems to make sense.

---

> ### Author Rebuttal · Authors · 2025-03-30
>
> Thanks for your reviews. We summarize your questions and provide our responses as follows:
>
> **References:**
>
> **R1: More discussions to Differential Privacy (DP) algorithms; Can we reduce one problem to another? For example if we split the privacy budget among $k$ layers, and apply exponential mechanism to them does it give us similar bounds?**
>
> Indeed, it is known that if an algorithm is $\beta$-differentially private, then its average sensitivity is at most $\beta$ (Varma & Yoshida, 2021). However, the reverse does not hold, as DP requires a bound on the worst-case sensitivity, whereas we can only bound sensitivity on average (i.e., under random deletions of points).
>
> It is unclear whether splitting the privacy budget across $k$ layers would yield a DP algorithm with similar bounds. Below, we highlight key obstacles in extending our result to the DP setting. Ensuring DP requires bounding the worst-case sensitivity, which in turn demands a non-trivial upper bound on the maximum cost of a deleted point to its assigned center, beyond the trivial bound of  $d\Lambda$. Unfortunately, we are unable to obtain such a bound. Instead, our analysis only gives a bound on average sensitivity— for instance, Lemmas D.2 and D.3 are based on the expected cost of a deleted point rather than the worst-case scenario.
>
> In future revisions, we will expand the discussion in the related work section to further explore the connection between DP and average sensitivity.
>
> **Q1: Can the methods here be applied to "flat" $k$-clusterings?**
>
> Please note that flat k-clustering is a sub-problem of hierarchical Euclidean k-median problem, in fact our algorithm solves the problem for all k at the same time.
>
> Thus, the methods can be applied to the flat k-clustering, but since our definition is stronger, so the results corresponding to average sensitivity of flat k-clustering might be slightly worse. In fact, Yoshida & Ito (2022) gave an approach for “flat” k-median clustering (and other Euclidean clustering), introducing a coreset-based method to achieve low average sensitivity. They showed that for the Euclidean $k$-median clustering algorithm with an $\alpha$-approximation, a coreset can be constructed with an average sensitivity of $\tilde{O}(\frac{dk^2}{\epsilon^3 n})$, yielding a clustering result that is a $(1+\epsilon)\alpha$-approximation with high probability.
>
> However, their notion of average sensitivity is defined with respect to total variation distance, thus their result is not completely comparable to ours. If we convert TV distance to our earth mover’s distance (EMD), which roughly involves multiplying by n, the resulting bound is $\tilde{O}(\frac{dk^2}{\epsilon^3})$. (Recall that our sensitivity bound is $O(k\ln n/\epsilon)$ and the approximation ratio is $d \log\Lambda (1+\epsilon)^k$).
>
> **Q2: The cost function seems insensitivity to relative differences among points. For example, if I take a cluster in the $k$ clusters and move it farther away the constructed tree still seems to be the same . I wonder if you have any opinion about what this means if we use the traditional cost functions for $k$-median.**
>
> In our definition, the $k$-median cost at each layer $k$ follows the traditional cost function for $k$-median. Intuitively, if a cluster in the optimal $k$-clustering is moved farther away, the same set of $k$  clusters should still be identifiable in the resulting dataset.
>
> However, our hierarchical clustering cost function is sensitive to the structure of the tree. Specifically, at any fixed layer $k$, moving a cluster (say,  $A$) farther away can significantly impact the clustering at layer  $k-1$. For example, in the original dataset, the algorithm might merge clusters $A$ and $B$, but after moving $A$ away, it may instead merge  $B$ with another cluster, $C$. This change in merging order can lead to structural differences in the hierarchy.
>
> **Q3: The paper starts with constructing the $2$-RHST tree. The hierarchical clustering then builds on the tree only and ignores the original datasets. Intuitively why do we choose the $2$-RHST tree? Are there other tree embeddings that can work for this problem?**
>
> As noted in the CLNSS paper, standard embedding techniques allow all points in the dataset to be embedded into a $2$-RHST with only a small distortion. Intuitively, the $2$-RHST effectively preserves the hierarchical structure of the original dataset and provides a natural way to construct a hierarchical clustering tree. This is achieved through a top-down approach that recursively partitions the point set based on distance.
>
> There are other tree embedding approaches for flat  $k$ -clustering. For example, Balcan et al. (2017) (Differentially Private Clustering in High-Dimensional Euclidean Spaces) introduced a tree partitioning method. However, it is unclear to us whether these tree embedding methods can be directly applied to the hierarchical  $k$ -median problem.

---

### Decision · Program_Chairs · 2025-05-01

**Decision:**

Accept (poster)

**Comment:**

All three reviewers recommend acceptance (weak accept). They agree that the paper addresses an interesting problem, hierarchical  k-median clustering under average sensitivity, and offers both novel theoretical insights and practical experiments. In particular, the reviewers commend (1) the theoretical analysis, which provides nontrivial average-sensitivity bounds and approximation guarantees, (2) the clarity of proofs, and (3) the empirical comparisons demonstrating the proposed algorithm’s robustness relative to standard linkage methods and a deterministic variant of CLNSS.

There were some concerns on novelty, but the merits outweigh the concerns in my opinion.